# Autonomous robotic searching and assembly of two-dimensional crystals to build van der Waals superlattices

Satoru Masubuchi[1], Masataka Morimoto[1], Sei Morikawa[1], Momoko Onodera[1], Yuta Asakawa[1], Kenji Watanabe [2], Takashi Taniguchi[2] & Tomoki Machida[1]

Van der Waals heterostructures are comprised of stacked atomically thin two-dimensional crystals and serve as novel materials providing unprecedented properties. However, the random natures in positions and shapes of exfoliated two-dimensional crystals have required the repetitive manual tasks of optical microscopy-based searching and mechanical transferring, thereby severely limiting the complexity of heterostructures. To solve the problem, here we develop a robotic system that searches exfoliated two-dimensional crystals and assembles them into superlattices inside the glovebox. The system can autonomously detect 400 monolayer graphene flakes per hour with a small error rate (<7%) and stack four cycles of the designated two-dimensional crystals per hour with few minutes of human intervention for each stack cycle. The system enabled fabrication of the superlattice consisting of 29 alternating layers of the graphene and the hexagonal boron nitride. This capacity provides a scalable approach for prototyping a variety of van der Waals superlattices.

[1] Institute of Industrial Science, University of Tokyo, 4-6-1 Komaba, Meguro Tokyo 153-8505, Japan. [2] National Institute for Materials Science, 1-1 Namiki, Tsukuba, Ibaraki 305-0044, Japan. Correspondence and requests for materials should be addressed to S.M. (email: mastoru@iis.u-tokyo.ac.jp) or to T.M. (email: tmachida@iis.u-tokyo.ac.jp)

The family of exfoliatable and functional two-dimensional (2D) crystals[1–4] is rapidly growing, exhibiting various electronic properties such as ferromagnets[5,6], semiconductors[7], superconductors[8], and topological insulators[9]. Recent advancements of 2D crystal research were enabled by key technological breakthroughs, including mechanical exfoliation and the assembly of van der Waals (vdW) heterostructures[1–3]. The precise interface controllability and wide choice of materials with various electronic properties enable vdW heterostructures to have high technological potentials that cannot be achieved by conventional semiconductor heterostructures[10–12]. Subsequently, technological improvements in visualization[13–16] and mechanical transfer[17–20] techniques have been implemented to meet the demands for various vdW heterostructures for fundamental scientific and device application research[10]. However, the basic manufacturing processes have relied on the manual operation of the experimental apparatus. In principle, the expected time needed to fabricate a vdW heterostructure consisting of $N$ layers scales as $(A + B)N \times (1 - p)^{-N}$, because the time needed for finding atomically thin 2D flake scales as $AN$, that for stacking the 2D crystals as $BN$, and the expected number of trials for successful fabrication is $E[p, N] = \sum_{k=1}^{\infty} k(1 - (1 - p)^N)^{k-1}(1 - p)^N = (1 - p)^{-N}$, where $p$ is the probability of failure per transfer. When we estimate $A = 20$ min, $B = 20$ min, and $p = 6\%$, which can be an optimistic value even for a well-trained researcher, the time required to complete $N = 29$ vdW heterostructure counts up to ~120 h of successive process. Performing such repetitive manual tasks is practically impossible.

Furthermore, the recent emergence of oxygen-sensitive functional 2D crystals, such as black phosphorous[21,22], niobium diselenide[23], and chromium triiodide[5], into the building block family of vdW heterostructures requires that the fabrication environment change from ambient conditions to the inert atmosphere enclosed in the glovebox. The reduced operability of the experimental apparatus in the glovebox enclosure further increases $A$, $B$, and $p$, thus limiting the complexity of vdW heterostructures within the feasible manufacturing steps. The most complicated vdW heterostructure reported to date is composed of 13 alternating layers of $MoS_2$, graphene, and hBN[24], which was attained after the days of manual operation[11]. Research on vdW heterostructures is currently at the crux of the increasing demand for higher complexity and the reduced fabrication feasibility. Obviously, a technological breakthrough is needed to solve the problem.

In the fields of biology, pharmacology, and chemistry, robotics and computer-vision technologies are now replacing manual experimental procedures, such as pipetting, specimen handling[25], and optical microscopy[26], thereby bolstering the scientific discovery potentials[27]. A key time has thus arrived for leveraging automation in the vdW heterostructure fabrication. Without automation, progress in vdW heterostructure research will stall, and the so-called dreamscape of designer materials comprised of vdW superlattices[11] will remain unrealized. Moreover, researchers will be encumbered by the endless tasks required for vdW heterostructure fabrication. To date, some technological components for automation have been proposed, such as image analysis algorithm for segmenting graphene flakes on $SiO_2/Si$[28], and layer by layer stacking techniques of CVD-grown 2D crystals[29]. However, the automated robotic fabrication of vdW heterostructures has yet been realized, which is presumably because of the high technological hurdles for integrating vdW heterostructure fabrication, computer vision, and robotic technologies.

Here, to solve the problem we report on the development of an autonomous robotic system for the assembly of vdW heterostructures, which we refer to as a 2D materials manufacturing system (2DMMS). The system was assembled in the glovebox enclosure with inert gas atmosphere. The system can autonomously detect 400 monolayer graphene flakes per hour with a small error rate (<7%) and stack four cycles of the designated two-dimensional crystals per hour with less than few minutes of human intervention for each cycle. The system enabled fabrication of the superlattice consisting of 29 alternating layers of the graphene and the hexagonal boron nitride. This capacity provides a scalable approach for prototyping a variety of van der Waals superlattices.

## Results

**The system architectures and functionalities**. Computer-assisted design (CAD) schematics of the proposed 2DMMS system are shown in Fig. 1a, and their photographs are presented in Fig. 1c–e. The functionalities of the system are schematically presented in Fig. 1b. First, the automated high-speed optical microscope with motorized XY scanning stage scans the surfaces of $SiO_2/Si$ chips, the optical microscope images are analyzed by computer-vision algorithm, and discerns whether targeted 2D crystals are present in the image or not (Supplementary Movie 1). When 2D crystals are detected, their positions and shape parameters are recorded in a database (Supplementary Movie 2), resulting in an immense catalog of available crystals. By choosing the crystals from the catalog using customized CAD software, the combination, relative positions, and crystallographic orientations of the 2D crystals are designed. Finally, the design is transferred to the stamping apparatus, and robots directed by the computer-vision algorithm assemble the vdW heterostructures layer by layer onto a polymer stamp through the vdW force (Supplementary Movies 3–5).

**Fabrication of van der Waals superlattices**. To demonstrate the fabrication of vdW superlattice structure using 2DMMS, we performed the following procedure. First, we exfoliated graphite and hBN crystals on centimeter-sized $SiO_2/Si$ chips. $SiO_2/Si$ chips[2,3] were loaded to the enclosure, tiled on the chiptrays and then scanned by the optical microscope. By searching 36 (11) $SiO_2/Si$ chips, we could detect 4600 (2000) graphene (hBN) flakes, whose positions and shape parameters are recorded in a database. Then, we choose the building block graphite and hBN flakes as shown in the left panels of Fig. 2a. The order, relative positions, and crystallographic orientations are designed simultaneously on the computer screen. The $SiO_2/Si$ chips were annealed in the Ar/ $H_2$ atmosphere at 600 °C for 3 h in the quartz furnace built inside the secondary glovebox connected to the main glovebox. Then we set the chiptrays to the stamping apparatus and ignited the assembly process. The robotic arm equipped with non-contact chuck transferred the $SiO_2/Si$ chip to the motorized XYZ stage (Supplementary Movie 3). The stage was heated to 80 °C. The optical microscope and the glass plate with polymer stamp was set over the $SiO_2/Si$ chips by the motorized linear stage. At this stage, the optical microscope is usually located away from the targeted 2D crystal flake. The optical microscope image was analyzed by the computer-vision algorithm, and XYZ stage was automatically navigated to align the targeted 2D crystals (Supplementary Movie 5). The $SiO_2/Si$ chip was lifted, and the targeted 2D crystal was contacted to the 2D crystal on the polymer stamp. The targeted 2D crystal adhered each other through the vdW force. The $SiO_2/Si$ chip was lowered, thereby picking up the targeted 2D crystal. The $SiO_2/Si$ chip was unloaded to the Si chip tray by the robot arm. The procedures between $SiO_2/Si$ chip loading and unloading were repeated until the designated vdW superlattice was formed.

The completed vdW superlattice was dropped onto $SiO_2/Si$ chip (the procedure is described in the Methods section). Finally,

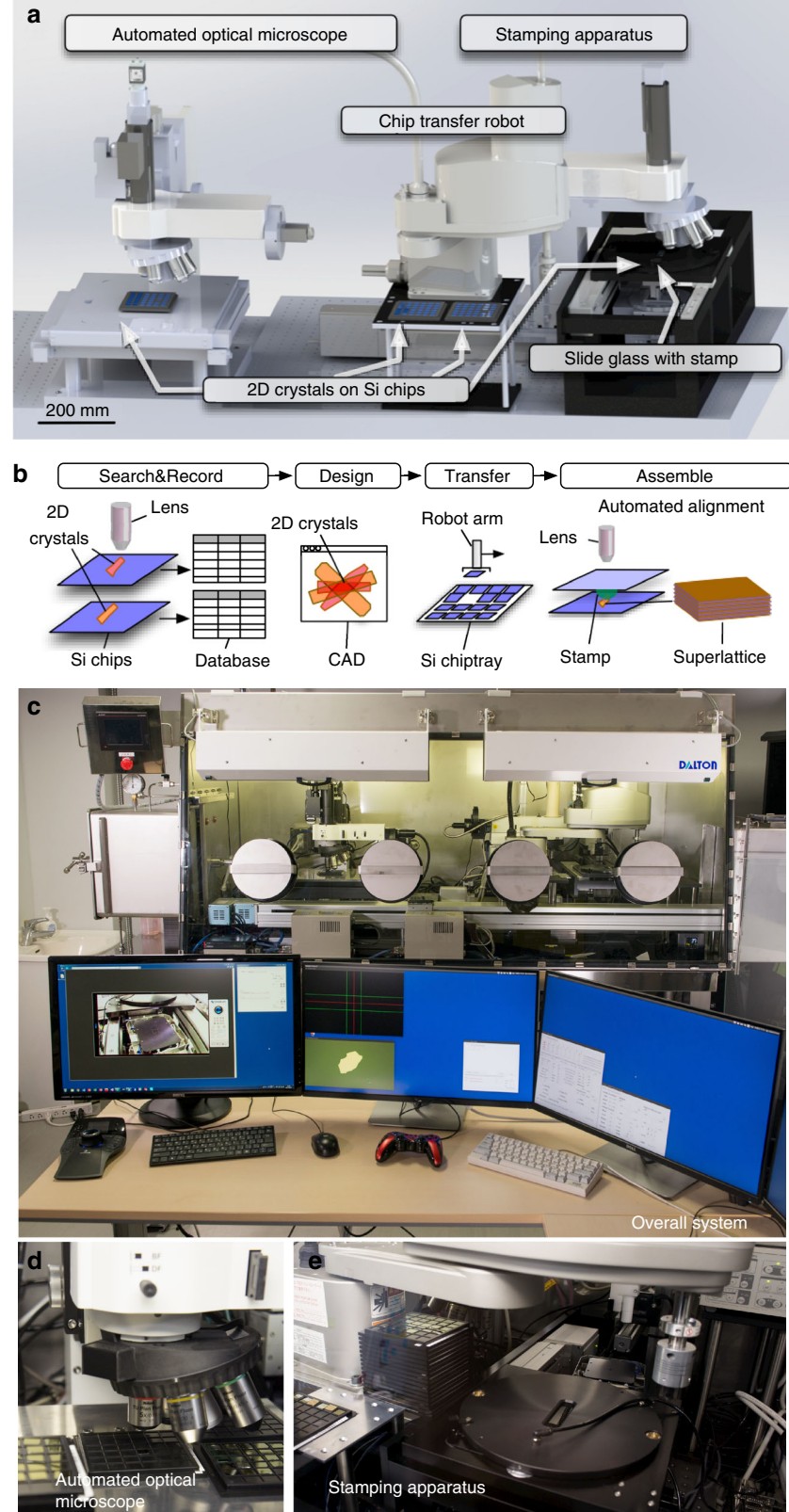

**Fig. 1** The computer-assisted design schematics, functionalities, and photographs of the system. **a** Computer-assisted design schematics of the presented robotic system comprising an automated optical microscope, stamping apparatus, and Si chip transfer robot. **b** Schematics of the vdW heterostructure fabrication process. First, the automated high-speed optical microscope with motorized XY scanning stage scans the surfaces of $SiO_2$/Si chips. When 2D crystals are detected, their positions and shape parameters are recorded in a database. By using customized CAD software, the combination, relative positions, and crystallographic orientations of the 2D crystals are designed. Finally, robots directed by the computer-vision algorithm assemble the vdW heterostructures layer by layer onto a polymer stamp through the vdW force. **c–e** Photographs of **c** entire system, **d** close up of automated optical microscope, and **e** stamping apparatus

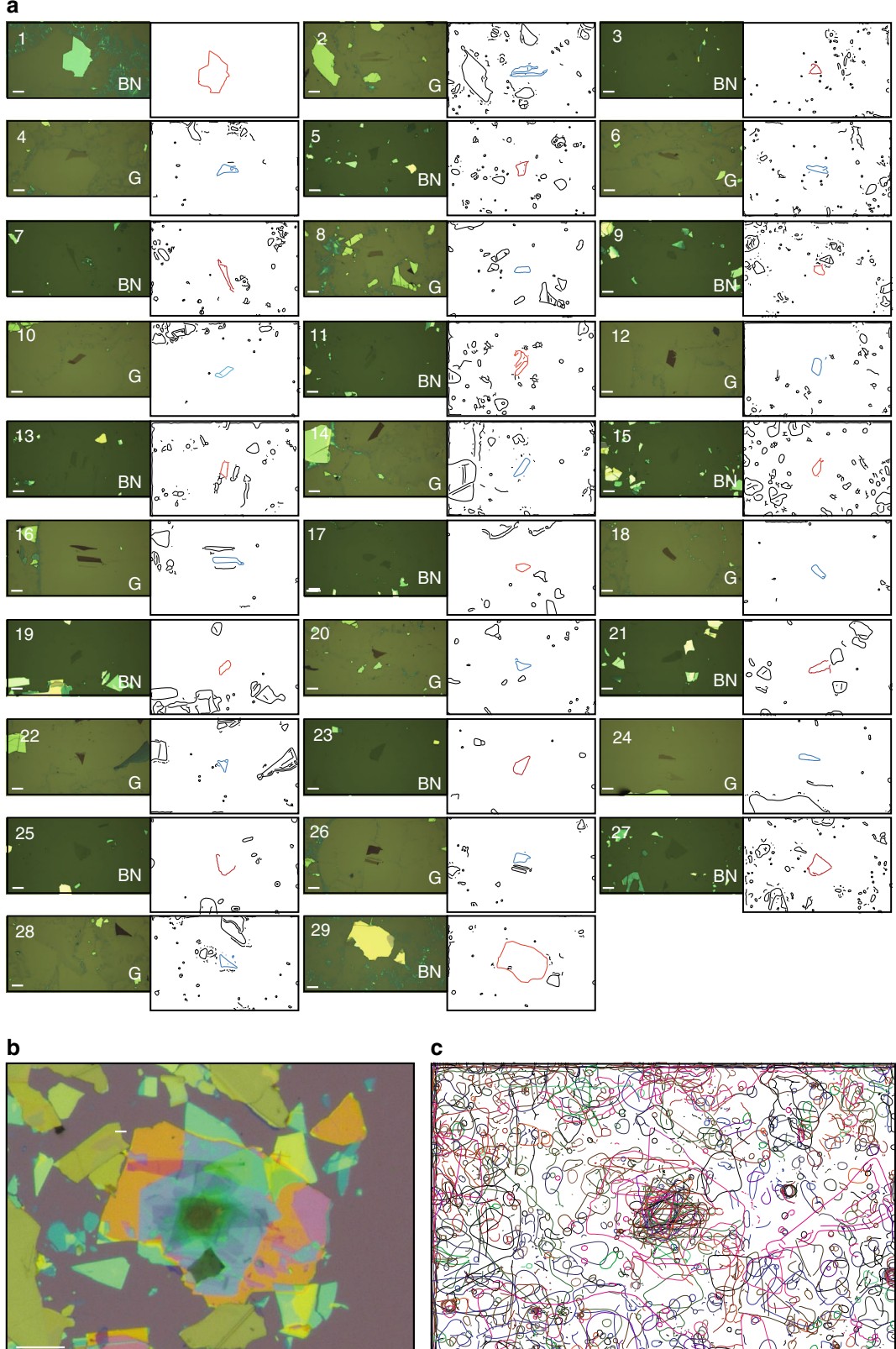

**Fig. 2** Van der Waals superlattices comprising 29 alternating layers of graphite and hBN. **a** Left panels: Optical microscope images of graphite (G) and hBN utilized to assemble vdW heterostructures. Right panels: The vector-drawing line-edges of G and hBN flakes extracted from the edge detection algorithms applied to the optical microscope images taken in the stamping apparatus before pick-up. The scale bars correspond to 20 μm. **b** Optical microscope image of G/hBN vdW superlattice structures. The scale bar corresponds to 20 μm. **c** Superimposed edges of G and hBN

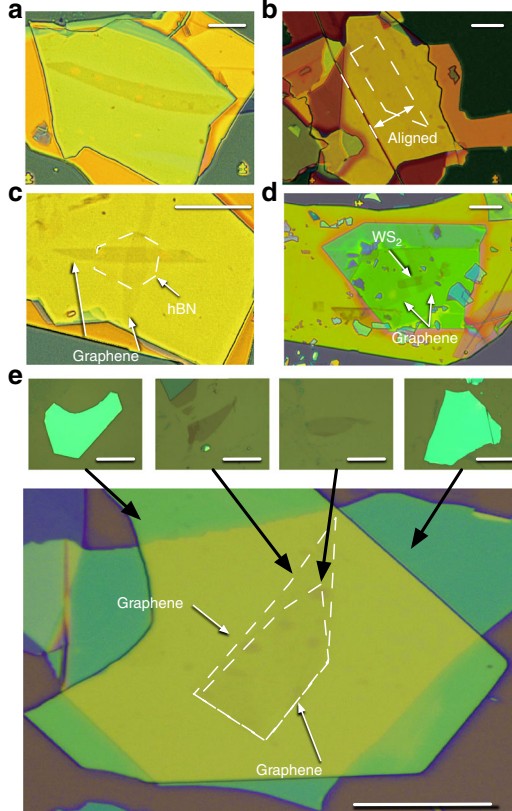

**Fig. 3** Optical microscope images of the vdW heterostructures fabricated by using the system. **a** Optical microscope images of trilayer graphene encapsulated in hBN. **b** Monolayer graphene with crystallographic axes aligned to those of hBN (indicated by white dashed lines). **c** hBN vertical tunneling structure. **d** hBN-encapsulated WS$_2$, which was contacted by trilayer graphene, where the scale bars represent 10 μm. **e** Bottom panel: Twisted monolayer–bilayer graphene encapsulated in hBN. Top panel, left to right: hBN, graphene, bilayer graphene, and hBN utilized to assemble the twisted monolayer–bilayer graphene, where the black arrows indicate the correspondence between images, and the white dashed lines indicate the graphene edges. The scale bar represents 10 μm

we obtained the vdW superlattice as shown in Fig. 2b. The right panels in Fig. 2a show the line-edges of G and hBN flakes extracted during the assembly process. The overlaid images (Fig. 2c) enable clarification of the positions of G and hBN flakes. The total machine operation time was less than 32 h, and the total human involvement for operation was less than 6 h (the breakdowns of the tasks are summarized in Supplementary Tables 1 and 2).

In addition, the system produced various vdW superlattices: trilayer graphene (Fig. 3a), a graphene/hBN Moiré superlattice (Fig. 3b), an hBN tunnel device (Fig. 3c), monolayer tungsten disulfide contacted by graphene (Fig. 3d), and twisted monolayer–bilayer graphene (Fig. 3e). The devices exhibited the highest transport and optical properties[19,30–34] (Supplementary Notes 1–4), and (Supplementary Figures 12–15), which demonstrated that the proposed system provides a practical approach for fabricating vdW superlattices[11].

**The points of developments**. To realize the above functionalities, two key developments were achieved: first, the automated microscopy, and second, the stamping apparatus, both of which were autonomously conducted by computer-vision algorithms.

The basic requirements in the automated microscopy were detecting atomically thin 2D crystals with a sufficiently small false detection rate, and the recording capability of the shapes and position parameters to comprise the searchable catalog. The basic requirement in the stamping apparatus was the automated addressing capability of the designated 2D crystals to the stamp. The use of metal alignment marks must be avoided for the SiO$_2$/Si substrates on which the starting 2D crystals were exfoliated because it contaminates surface and restricts the thermal cleaning capability[35]. On the other hand, the fabricated vdW heterostructure should be transferred onto the SiO$_2$/Si with metal alignment marks for further processing.

We fulfilled the requirements by implementing the following innovations: first, the image processing pipeline to distinguish 2D crystals that are robust to the contaminating material that forms in the exfoliation processes. Second, the global positioning system of the exfoliated 2D crystals without use of pre-patterned alignment masks. And third, the integrated information system that runs these algorithms on our developed robotic system.

**Image processing pipeline to distinguish 2D crystals**. In Fig. 4a, we present the developed image processing pipeline to distinguish 2D crystals. The figure additionally shows the optical microscope image of monolayer graphene on SiO$_2$ (Fig. 4b) and the outputs of the algorithms (Fig. 4c–h). The pipeline was composed of the double-filtering process based on the color contrast and the information entropy. This was necessary because the simple application of the previously reported algorithm[28] to the microscope images resulted in a small percentage of true detection (<50%), which was derived mainly from the contaminating materials of colors similar to those of atomically thin 2D crystals (Supplementary Note 5 and Supplementary Figure 16).

First, the image processing pipeline was split into the color and entropy thresholding (Fig. 4a). In color thresholding, we apply the function to the acquired image $I^{H,S,V}(x, y)$ as

$$f_{\text{thresh}}(x,y) = \begin{cases} 1, & \begin{aligned} C^{H,S,V} - D^{H,S,V}/2 \leq I^{H,S,V}(x,y) \\ -I_{\text{BG}}^{H,S,V}(x,y) \leq C^{H,S,V} + D^{H,S,V}/2 \end{aligned}, \\ 0, & \text{otherwise} \end{cases} \quad \text{where}$$

$I_{\text{BG}}^{H,S,V}(x,y)$ is the background image of SiO$_2$/Si without 2D crystals (recorded manually before starting the automated search), and $C^{H,S,V}$ and $D^{H,S,V}$ are the parameters (Supplementary Table 3) adjusted by the procedure described in the Methods section. This procedure extracts the region of atomically thin 2D crystals (Fig. 4g). However, the regions other than targeted crystals—corrugated crystals and scotch tape residues—remain in the segmented image (white arrows in Fig. 4g). Then, we apply entropy thresholding. To depict the effects, Fig. 3i shows the local information entropy $H$ calculated with a mask size of $9 \times 9$ (pixels)$^2$ from the image in Fig. 3c. The line cuts of Fig. 3i at $Y = 200, 400,$ and 600 pixels are shown in Fig. 4j. As indicated by the black arrows in Fig. 3i, j, the regions of contamination, as described above, have a larger $H$ than those of exfoliated 2D crystal flakes.

Therefore, to eliminate the regions with a large $H$, we assembled the pipeline using the following procedure. First, edge detection was applied to a grayscale image (Figs. 4c, d). The morphology operations were then applied to fill the regions surrounded by the edges (Fig. 4e). We then calculated the entropy $H = -\Sigma_{i=0}^{255} P_i \log_2 P_i$ of the grayscale image for each region, where $P_i$ is the gray value histogram of the pixels which belong to the region. By selecting regions with smaller entropy than the threshold value $H \leq U_{\text{Entropy}}$, the regions with contaminating objects were eliminated (Fig. 4f, Supplementary Notes 5 and 6) and (Supplementary Tables 4–6). Finally, by taking the intersection between the color-threshold and entropy-threshold regions

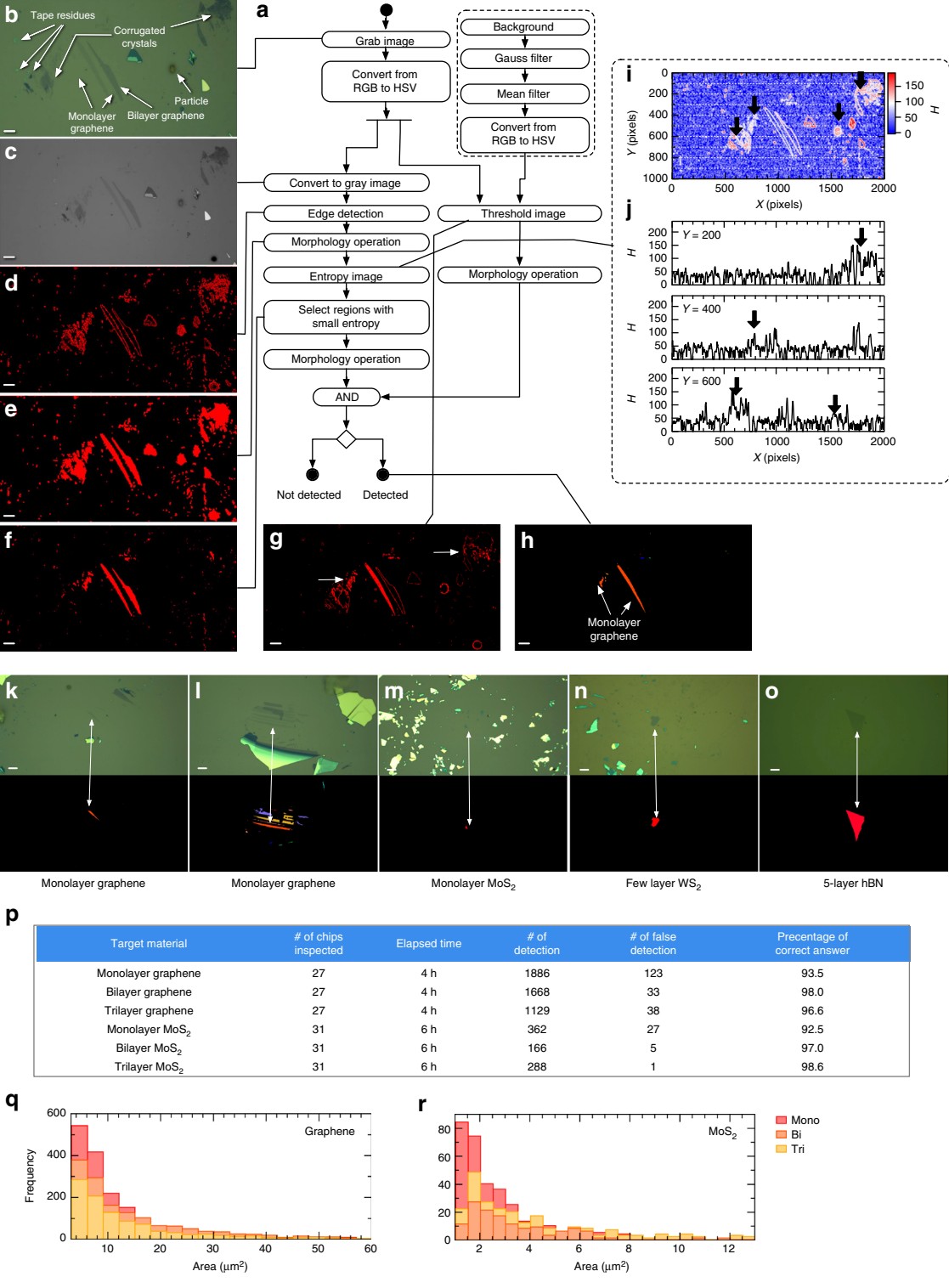

**Fig. 4** The pipeline and results of image processing algorithm. **a** Schematic of processing algorithm for detecting atomically thin 2D crystals from optical microscope images. **b** Optical microscope image of exfoliated graphene flakes on $SiO_2$/Si chips, where the scale bar represents 10 μm. The image includes: monolayer graphene, bilayer graphene, Scotch tape residues, collapsed flakes, and Si particles. **c–h** Representative images generated at each step. The correspondence between images and blocks are indicated by solid lines. The scale bars represent 10 μm. **i** Entropy $H$ extracted from the optical microscope image in **c** with a mask size of 9 × 9 pixels. **j** Line cuts of **i** at $Y = 200$, 400, and 600 pixels (from top to bottom). The thick black arrows indicate the positions of particles, Scotch tape residues, and corrugated graphene flakes. Results of the detection algorithm presented applied to **k**, **l** monolayer graphene, **m** monolayer $MoS_2$, **n** a few layers of $WS_2$, and **o** five-layer hBN. The scale bars represent 10 μm. The top panels show the input optical microscope images; the bottom panels show regions extracted as 2D crystals. **p** Performance metrics for the proposed automated searching system. **q**, **r** Size distributions of exfoliated graphene (**q**) and $MoS_2$ (**r**)

(Fig. 4h), we could extract the region containing targeted 2D crystals from the optical microscope image. When any regions existed with areas larger than threshold $L_{area} \leq S(n)$, the pipeline discerned that 2D crystal flakes were present in the acquired image.

**Results and performance metrics of automated searching system.** After equipping the automated microscope system with the presented pipeline, we conducted automated searching for graphene and $MoS_2$. The performance metrics are presented in Fig. 4p. The system analyzed 12,000 images per hour and identified more than 400 monolayer graphene flakes per hour, with a sufficiently low false detection rate of <7%, while distinguishing the layer numbers. To demonstrate the proposed algorithm's capability of detecting 2D crystals, in Fig. 4k–o we show the representative outputs of its detection (bottom rows) applied to monolayer graphene (Fig. 4k, l), monolayer $MoS_2$ (Fig. 4m), a few layers of $WS_2$ (Fig. 4n), and five-layer hBN (Fig. 4o). Note that the regions of the targeted 2D crystals are successfully partitioned, while the contaminating objects are removed. Thus, the system is also applicable for finding $MoS_2$ flakes (Fig. 4p).

Owing to the above feature, we could collect big data, including the size distribution of exfoliated crystals, as presented in Fig. 4q, r. The acquired information (optical microscope images, positions of the XY stage and objective lens, and shape features) were recorded in the database. Finally, the system captured the optical microscope images of the detected 2D crystals with objective lenses of various magnifications (×20, ×10, and ×5), which were later utilized to locate 2D crystals in the stamping process.

**CAD of van der Waals superlattices.** When the automated searching process is completed, the database comprises a large list of exfoliated 2D crystals on $SiO_2$/Si, which is sortable by the arbitrary extracted features, such as the area, rectangularity, anisotropy, and compactness[36]. Therefore, one can select the favorable 2D crystals from the catalog of 2D crystals. By stacking the previewed 2D crystals (Fig. 5c) on each other (Fig. 5b, d), the vdW superlattice is designed. In this step, the order, relative angles, and positions of the 2D crystals are designed, which is assisted by dynamic edge extraction (Fig. 5d) and the edge angle fittings (Fig. 5e, and Supplementary Figure 19). The vdW superlattices presented in Fig. 2b and Fig. 3 were designed in this way.

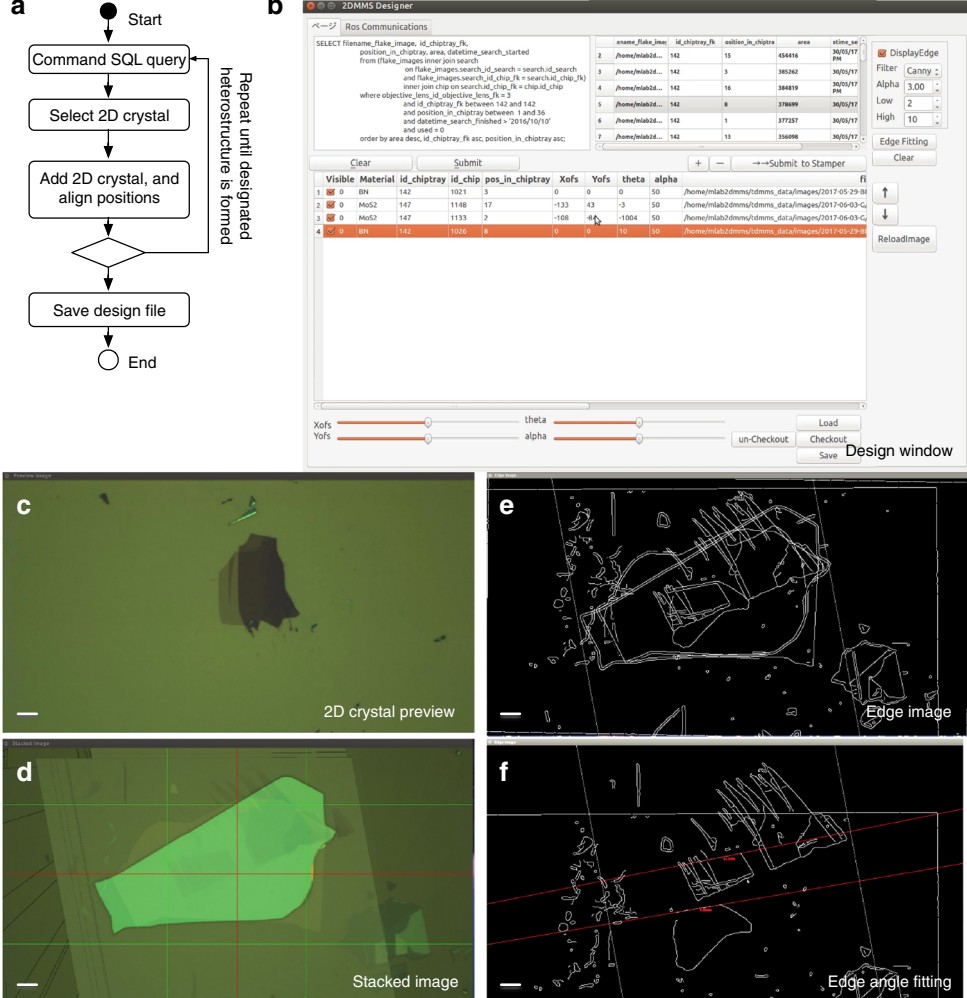

**Fig. 5** The computer-assisted design of vdW heterostructures. **a** Operational flow diagram for designing vdW heterostructures. **b–f** Graphical user interface (GUI) window screenshots of our CAD software. **b** Main control window. **c** 2D crystal preview. **d** Edge image. **e** Stacked image. **f** Edge angle fitting. The scale bars represent 10 μm

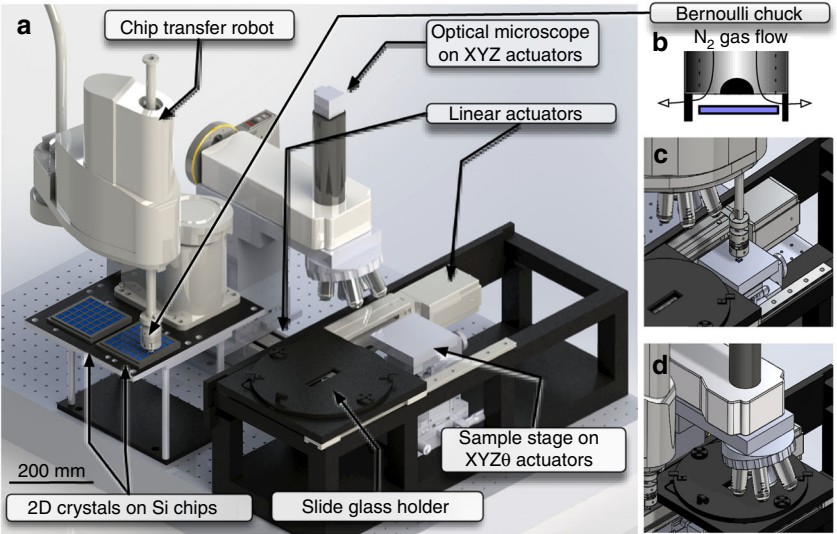

**Fig. 6** The automated stamping system. **a** Schematic of automated stamping system. **b** Schematics of robot arm end effector; Bernoulli-type floating chuck. **c**, **d** Schematics of the automated stamping procedure: transferring $SiO_2$/Si chips by robotic arm and setting optical microscope, respectively

**Assembly of van der Waals superlattices.** After completing the vdW superlattice design, the component 2D crystals were assembled by the automated stamping system using a polymer stamp on a glass slide (Fig. 6a). At this stage, the essential problem we had to solve was to establish a reliable alignment method of the targeted 2D crystals. In general, after loading the $SiO_2$/Si chip to the stamping apparatus, the optical microscope is usually positioned away from the targeted 2D flake. In the manual operation, locating small-sized 2D crystals (typically $10 \times 10 \ \mu m^2$) in large $SiO_2$/Si chips (typically $10 \times 10 \ mm^2$) is a remarkably challenging task. In particular, in the case when $SiO_2$/Si chips are rotated from the original angle, which often becomes the case for aligning the crystallographic angles of 2D crystals, the diminished cognitive ability of the operators results in exponentially increased times needed for the alignment.

Without implementing an automated alignment, the fabrication of superlattice structures (Fig. 2b) is not feasible. Based on this background, we developed the global positioning system of 2D crystals on $SiO_2$/Si substrates. The key idea was utilization of the shapes of the randomly distributed 2D crystals as a unique fingerprint pattern identifying the current position of the optical microscope. Accordingly, we leveraged the weak point of the mechanical exfoliation process—randomness—as an advantage in the positioning.

The alignment procedure is schematically presented in Fig. 7a–c. First, we capture the optical microscope image at the present position and extract the edges of the 2D crystals. The extracted edge patterns are matched with those stored in the database (Fig. 7a). Since the template images are recorded with their position information $\left(x^i_{temp}, y^i_{temp}\right)$, we can calculate the offset from the current position to the targeted 2D flake $\left(x^i_{target}, y^i_{target}\right)$ from the results of matching (I, and θ) (Fig. 7b). The representative results of matching are presented in (Fig. 7d–e).

After completing the above process, we move the optical microscope to the targeted 2D crystal flake within a lateral error of ~300 μm. This procedure enables rapid addressing of targeted 2D crystals within 1 min (Supplementary Movies 4 and 5). When the automated alignment process is completed (Fig. 7c), the targeted 2D crystal is brought to the center of the optical

microscope within a lateral error of 10 μm (Fig. 7f–i, Supplementary Figure 20, and Supplementary Figure 21). The human operator is requested to confirm the alignment position before the process proceeds further. If it is needed, the operator can manually override the alignment and perform fine tuning using the alignment assistance tools presented on the computer screen (described in the Methods section). Then, the 2D crystal is lifted by applying a polymer stamp (Fig. 7j, k). This process is repeated by changing $SiO_2$/Si chips (Supplementary Movie 3). To avoid manually handling the centimeter-sized $SiO_2$/Si chips in the glovebox, $SiO_2$/Si chips are changed with a programmed robotic arm equipped with a Bernoulli-type floating chuck with $N_2$ gas flow, which prevents mechanical distortion of the 2D crystal $SiO_2$/Si chip (Supplementary Movie 3).

## Discussion

The fabrications of the vdW superlattice structures presented in Figs. 2 and 3 was achieved by implementing the above technologies. Here, we would like to discuss the current limitations of the setup as well as the outlook for the future improvements. These can be split into three technological groups: the stamping process, the alignment, and the human intervention. First, in the stamping process, the formation of blisters between 2D crystals are not fully avoided (Supplementary Note 4). To solve the problem, the improvements in the stamping process, such as the hot pick-up method[37], needs to be implemented. Second, the final alignment accuracy gained after using the alignment assistance tool becomes less than ±1° and ±1 μm, respectively (Fig. 7). However, to achieve this accuracy, manual alignment process is needed (Supplementary Notes 8–11). This process needs to be automated in the future developments of the image detection algorithm and error handling schemes. Third, for the human intervention, the exfoliation and final transfer was performed manually outside the glovebox. To automate these procedures, robotic systems for exfoliation, wafer cleaning, and Si chip tray transfer between the components, etc. will be needed. The recent rapid developments in the machine learning and computer-vision algorithms might help realizing these technologies.

In summary, we developed the autonomous robotic system for searching 2D crystals and assembling them into vdW heterostructures. Because this system is contained in a glovebox, the

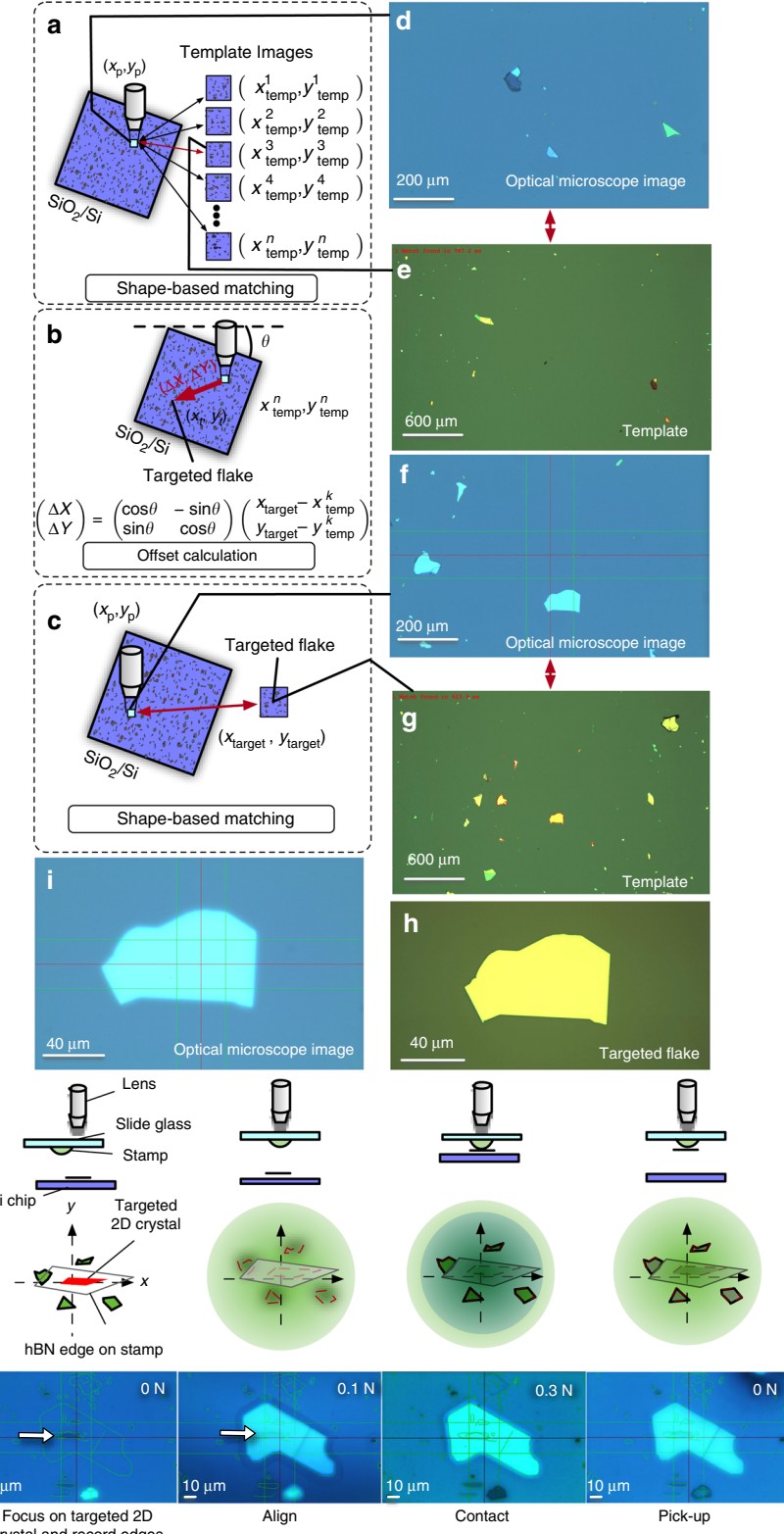

**Fig. 7** Alignment procedure. **a** Schematic of the matching process conducted to extract the current position of the optical microscope $(x_p, y_p)$ from the captured image (square filled with light blue). **b** Geometrical relationship between the targeted flake $(x_{target}, y_{target})$ and the extracted position of the optical microscope $\left(x_{temp}^k, y_{temp}^k\right)$. **c** Schematic procedure for aligning the targeted 2D crystal to the optical microscope center. **d**–**e** Representative sample of shape-based matching between the optical microscope image (**d**) and the template image (**e**). The matched 2D crystal edges are indicated by red curves in **e**. **f**, **g** Representative sample of shape-based matching between the optical microscope image (**f**) and the template image (**g**). **h** Optical microscope image after automated alignment and **i** template image of the targeted 2D crystal. **j** Schematics of the alignment procedure to lift targeted 2D crystals. **k** Optical microscope images obtained at each step of **j**

system can handle oxygen- and humidity-sensitive 2D crystals, such as black phosphorous[21,22] and niobium diselenide[23]. In addition, the system is in principle applicable to the assembly of heterostructures from CVD-grown 2D crystals (Supplementary Note 12). The wider material design freedom enabled by our system offers unprecedented opportunities for exploring the full potential of vdW heterostructures. By this development, we can reduce the human intervention involved in the vdW heterostructure fabrication by orders. We believe that this work free up researchers from repetitive tasks and letting them to focus on more intellectually creative tasks. Therefore, this is a fundamental step forward to realizing the dreamscape of artificial materials by vdW heterostructures. With respect to the materials aspect of vdW heterostructures, the digital nature of the materials properties of 2D crystals and the wide choice of combinations for vdW heterostructures have good compatibility with the information-processing algorithms of machine learning and Bayesian statics[38], which will open new trajectories of combinatorial materials research on vdW superlattices.

## Methods

**Hardware and software design**. Because no manufacturer provides the proposed system in its entirety, we were required to integrate (Supplementary Figure 1) a large number of hardware components (Supplementary Figure 2) and develop the software system (Supplementary Figure 3). The hardware units involved in this system are connected to a Linux workstation via conventional communication interfaces (USB, RS-232C, camera link, and Ethernet).

Our customized software system was written in C++ and Python. The source code is publically available under an open source license at the code repository (https://github.com/tdmms/tdmms). The researchers interested in this work can download the source code from the GitHub and they can compile the code on their own computers. In addition, the researchers can join the software development using the social interfaces of the code repository.

The software runs with the aid of software frameworks from the Robotic Operating System[39], which provides a flexible inter-node messaging framework, enabling modular-based software assembly, which supports our system's robustness against changes in algorithms, hardware components, and conflicts between human and computer commands. The graphical user interface (GUI) was constructed using the QT4.0 GUI toolkit. The GUI was carefully designed to minimize human intervention so that the searching, design, and assembly of 2D crystals are executed with only a few clicks. Image processing algorithms were built on the HALCON framework (MVTec Software GmbH) and Open CV.

**Connectivity of the hardware components**. In order to provide the information on the hardware connectivity, we present in Supplementary Figure 2 the connectivity diagram between the hardware components utilized in the system. This schematic describes the complete logical connection between the hardware components. By replicating the hardware connection and installing the software, the other researchers can reproduce the functionalities presented in this paper. However, at the same time, we must admit that providing the full and concrete description of the hardware system is virtually impossible because the system includes many hand-crafted parts. Therefore, we would like to ask the researchers who are interested in replication of this system to contact the author directly.

**Automated searching system (2DMMS-Finder)**. Supplementary Figure 4A shows the CAD schematics of our automated optical microscope system. The system is composed of a high-speed XY scanning stage, automated focusing system, and CMOS camera. The system can capture optical microscope images of SiO$_2$/Si chips every 80 ms, while moving in the XY stage. Supplementary Figure 4B schematizes the system's operational flow diagram. First, the entire region of an SiO$_2$/Si chip tray is scanned with an objective lens of the smallest magnification (×5), and our image processing algorithm extracts the positions of SiO$_2$/Si chips, as detailed in the following section.

Second, the system records alignment images, which are later utilized to restore the positions of the 2D crystals in the stamping apparatus. Third, the objective lens is switched to a higher magnification (×50), and optical microscope images are captured while incrementally moving the SiO$_2$/Si chips. At each step, our image processing algorithm analyzes the optical microscope images and determines the presence of the targeted 2D crystals. When a 2D crystal is detected, the detected region is moved to the center of the optical microscope.

The program then checks whether there are duplicate data in the relational database according to the position of the XY stage. If no duplicate data are found, the information (optical microscope images, positions of the XY stage and objective lens, and shape features) are recorded in the database. By this process, we avoid generating duplicate records of the same 2D crystal. Finally, the optical microscope

images of detected 2D crystals are recorded with objective lenses of various magnifications (×20, ×10, and ×5). The information is later utilized to locate 2D crystals in the 2DMMS-Stamper.

**Method of tuning the thresholding parameters in 2DMMS-Finder**. In Supplementary Figure 5A, a representative optical microscope image of monolayer and bilayer graphene on SiO$_2$/Si is shown. The values of the hue-saturation-value channels of Supplementary Figure 5A are shown in Supplementary Figure 5B–D, respectively, and the line cuts of each channel are shown in Supplementary Figure 5E–G, respectively. From this plot, the parameters utilized in Supplementary Figure 5 for extracting monolayer graphene and bilayer graphene are selected as $(C_H, D_H; C_S, D_S; C_V, D_V) = (0, 10; -8, 8; -10, 4)$ and $(0, 10; -14, 8; -19, 4)$, respectively (Supplementary Table 3).

**Automated extraction algorithm for the positions of SiO$_2$/Si chips in 2DMMS-Finder**. Supplementary Figure 6A shows schematics for the image processing algorithm used to extract the number and positions of SiO$_2$/Si chips. The XY stage is incrementally moved, and the images are acquired. The acquired images are then converted to grayscale images as $I_{gray}(x,y) = 0.299I_R(x,y) + 0.587I_G(x,y) + 0.114I_B(x,y)$, where $I_R$, $I_G$, and $I_B$ are the intensities of the red, green, and blue channels, respectively. The acquired images are then tiled into a single image (Supplementary Figure 6B), and a threshold is applied:

$$f(x,y) = \begin{cases} 1, & 50 \le I(x,y) \\ 0, & \text{otherwise} \end{cases}$$

(Supplementary Figure 6C). Morphology operations (opening/closing) are applied to $f(x,y)$ to eliminate the irrelevant particles, and the number of remaining regions is considered to be the number of SiO$_2$/Si chips. Rectangles are then fitted to each region (Supplementary Figure 6D), and regions surrounded by four vertex points of a rectangle are defined as the regions to be searched.

**Automated assembly system (2DMMS-Stamper)**. Supplementary Figure 7A shows CAD schematics of our automated stamping system. The operational flow chart and schematics of the stamping system in operation are presented in Supplementary Figure 7B and Supplementary Figure 7C–F, respectively. First, the optical microscope and glass slide holder are positioned away from the stamping stage (Supplementary Figure 7C). The robot transfers the SiO$_2$/Si from the chip tray to the stamping stage (Supplementary Figure 7). The robot returns to the original position (Supplementary Figure 7E), and the optical microscope and glass slide holder slide over the Si chip (Supplementary Figure 7F).

**Database structure**. To manage the data generated by the automated searching system, we developed a relational database system on MySQL, the structure of which is schematized in Supplementary Figure 8. We assigned a unique identification number to each Si chip and chip tray, referred to as a "Chip ID" and "Chip tray-ID," respectively. The positions of the Si chips in the chiptrays were recorded in the field "Position-in-chip tray." Each time a search began, a new entity was added to the table ('search action'). When the targeted 2D crystals were found, the entities were added to the table ('images of flakes'). The entities comprise a position of the XY stage (x and y) and the status of the optical microscope ('optical microscope parameters'). In addition, certain experimental parameters (type of Si wafer, crystal characteristics, and the name of the operator who executed the mechanical exfoliation) were simultaneously stored in the database (in 'Exfoliation, 'Operator', 'Parameters', and 'Materials' fields). These tables were accessed through the standard SQL queries, thus enabling unified management of the information generated during the fabrication process.

**Fabrication procedure of polymer stamp on optical microscope slide**. A thermoplastic methacrylate copolymer (Elvacite 2552C, Lucite International) powder was dissolved in anisole with a volume ratio of 1:1. A small droplet of the Elvacite/anisole solution was placed on the optical microscope slide using a toothpick. The optical microscope slide was baked on a hotplate at 130 °C for 5 min to evaporate the anisole solvent.

**Transfer process of vdW heterostructures onto SiO$_2$/Si**. To complement the information on the device fabrication, we added the pictorial flow diagram on the overall process of the device fabrication, especially focusing on how we handled the polymer stamp in Supplementary Figure 9.

When the alignment was complete, the polymer stamp was pressed onto the 2D flake (Supplementary Figure 9D). During the stamping, the sample stage was heated to 80 °C. The force acting between the polymer stamp and the Si chips was monitored by a load cell unit. Maintaining the contact force below the predetermined threshold value of $f = 300$ mN avoided fracturing the 2D crystals. After a duration of 20 s to promote adhesion between the 2D crystals, the polymer stamp was slowly lifted, thereby picking up the 2D crystals by van der Waals force acting between 2D crystals. This procedure was repeated until the desired vdW heterostructures were formed on the polymer stamp. (We discussed the typical cases which we fail in pick-up in Supplementary Note 7, Supplementary Figures 17, and 18.)

After finishing the vdW heterostructures assembly, the glass slide was unloaded from the stamping apparatus, taken out of the glovebox (Supplementary Figure 9E). The final transfer was conducted by using the similar stamping apparatus developed in the ambient air. The SiO$_2$/Si chip was cleaned by the piranha solution (H$_2$SO$_4$:H$_2$O$_2$ = 2:1), followed by the ultrapure water rinse. The SiO$_2$/Si was loaded to the stamping apparatus and was heated to 185 °C. The glass slide was set to the stamping apparatus. The position where the vdW heterostructure was dropped was selected by using the XY stage. The SiO$_2$/Si chip was slowly lifted and contacted to the vdW heterostructure. Then the polymer stamp was melted, thereby transferring the vdW heterostructure to the SiO$_2$/Si chip. After a short duration of ~20 s, the SiO$_2$/Si chip was lowered. The polymer stamp was prolonged and torn. Finally, the SiO$_2$/Si chip was unloaded from the stamping apparatus and the stamp was dissolved by immersing SiO$_2$/Si chip in chloroform for 1 min.

**How to control the vdW stack is peeled onto the Si substrate and not on the polymer stamp**. Whether the vdW stack is peeled to the SiO$_2$/Si substrate or stuck on the polymer stamp is controlled by the substrate temperature. A threshold temperature is determined by the glass transition temperature of the polymer utilized; $T_g$ = 90 °C for the case of Elvacite 2552C (Technical Data Sheet of Elvacite 2552 C). When the substrate temperature is lower than the glass transition temperature $T$ = 80 °C < $T_g$, the stack adheres to the polymer stamp. When the substrate temperature is higher than the glass transition temperature $T$ = 185 °C < $T_g$, the polymer stamp is melted, and the stack is transferred to the SiO$_2$/Si substrate.

**Automated extraction of edge patterns during final alignment**. While stamping, the targeted 2D crystals had to be envisaged through the glass slide and the polymer stamp. Although the polymer stamp was transparent, the edges of the 2D crystals were obscured by light scattering and refraction at the polymer stamp surface (second panel of Fig. 7k). To resolve this problem, we implemented a procedure to aid the alignment (Fig. 7j).

First, the edges of the 2D crystals were automatically extracted by applying Canny's edge detection algorithm. At this point, the edges of targeted 2D crystals as well as other thicker 2D crystals were extracted. The extracted edges were saved in vector-drawing format. The recorded edges were then overlaid on the optical microscope image on the computer screen (green curves in Fig. 7k). Then, the optical microscope was aligned with the Elvacite stamp. The targeted 2D crystal was observed with the overlaid edges (second panel of Fig. 7k). Note that these procedures were implemented in the software system and thus can be utilized by few clicks on the computer screen.

Now, although the edges of the targeted 2D crystals were obscured by the presence of the Elvacite stamp, the presence of thicker flakes scattered around the targeted flake exhibited a stronger optical contrast that could be aligned with the overlaid edge patterns. In the current form of the software implementation, this alignment was conducted by the operator. Following this procedure, we could achieve alignment of targeted 2D crystals within a lateral error of 1 μm (third panel of Fig. 7k). During the stamping process, the vertices of 2D crystals were recorded as a vector-drawing file to record the crystal positions. This information was helpful for identifying thin layers of 2D crystals, which became transparent when embedded in the encapsulating hBN flakes. Owing to these features, we could reliably design mesa-pattern and metal-contact shapes for the devices.

**Fabrication method of metal electrodes to vdW heterostructures**. To fabricate the metal electrodes, we spin-coated a 10:18 solution of polymethyl methacrylate (PMMA)-A6:anisole at 4,000 rpm, and the chip was baked at 180 °C for 20 min in an oven. Hydrogen silsesquioxane (HSQ; Dow Corning Fox-12) was then spin-coated at 4000 rpm, and the chip was baked at 120 °C for 2.5 min in an oven. The mesa pattern, developed by NMD-3 (Tokyo Oka Kogyo Co., Ltd), was written by electron-beam lithography (EBL) with a dose of 450 μC/cm$^2$. The chip was etched with an inductively-coupled plasma etching system (SAMCO, Inc.) using O$_2$ and CF$_4$/Ar plasma[19]. To produce the metal-contact pattern, a 3:2 solution of PMMA-A6:anisole was spin-coated at 4500 rpm, and the chip was baked at 180 °C for 30 min in an oven. The contact pattern was written by EBL with a dose of 400 μC/cm$^2$. The pattern was developed in a 3:1 solution of isopropanol (IPA) and methyl isobutyl ketone (MIBK) for 30 s, followed by a 2-min rinse with IPA. Au/Pd/Cr (45/15/10 nm) were deposited by electron-beam evaporation. The metal layer was lifted off by acetone.

**Validation process of the correct and false detection rates**. We describe the validation process we took to extract the correct and false detection rates in Fig. 4 in the main text. First, we note that the correlation between the optical color contrast and the layer thickness has been verified by using the transport and photoluminescence measurements. Supplementary Figure 10 shows the quantum Hall effects measured in (A) monolayer, (B) bilayer, and (C) trilayer graphene. Supplementary Figure 11 shows the Raman spectrum of monolayer, bilayer, and trilayer MoS$_2$. From which we assigned the layer thickness. Furthermore, the thickness of SiO$_2$ was controlled to 290 ± 5%, and the light illumination condition was kept constant by using the halogen lamp with the feedback control (LA-150FBU, Hayashi Watch Works). Therefore, the correlation between the optical

microscope images and the layer thickness is tightly controlled. Upon these basis, the false detection rate was manually aggregated by checking all the extracted optical microscope images.

Supplementary Table 5 shows the spreadsheet summary utilized for calculating the false detection rate. Note that the confusion between layer thickness is relatively small and the major source of false detection was the extraction of contaminating objects such as particles (Supplementary Note 5).

**Data availability**. The data that support the findings of this study are available from the corresponding author upon request. Readers interested in the detailed information on the hardware and software implementation should please contact S.M.

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

## Acknowledgements

The authors would like to thank Y. Hoshi, Z. Dou, M. Arai, and R. Moriya for helpful discussions and technical assistance. S.M. would like to thank T. Kitahara for providing the opportunity for gaining the experiences in autonomous robotics programming in his science, technology, engineering, and mathematics (STEM) education program. This work was supported by the Core Research for Evolutional Science and Technology (JPMJCR15F3), the Japan Science and Technology Agency (JST), and JSPS KAKENHI (Grant Nos. JP16H00982, JP25107003, and JP25107004).

## Author contributions

S.M. designed 2DMMS, implemented the hardware and software, and fabricated the vdW heterostructures. M.M. performed mechanical exfoliation of the 2D materials and analyzed the results of automated scanning for 2D material flakes. S.M., M.O., and Y.A. fabricated devices from the assembled vdW heterostructures and conducted transport measurements. K.W. and T.T. synthesized the hBN crystals. T.M. supervised the research program. All authors wrote the paper.

## Additional information

**Competing interests:** The authors declare no competing interests.

