## [Peer Review File(PDF 5302 kb) · Nature Communications]

Reviewers' comments:

Reviewer #1 (Remarks to the Author):

Masubuchi et al report on the development of a novel automated protocol and apparatus for micromechanical stacking of van der Waals heterostructures. The work is of high quality, definitely novel and very timely, which allows me to recommend the paper for publication in Nature Communication after the following comments are addressed:

1. The details of the heterostructure fabrication are rather vague. How often the polymer stamp has to be replaced? How was it done? How was the final transfer onto the silicon substrate made? Was it automated as well? Also, how to reliably control that the stack can be peeled onto the Si substrate and not stuck on the polymer stamp? How often was the human intervention needed during the van der Waals assembly?
2. How clean are the interfaces between the 2D crystals in the fabricated stacks? AFM images for graphene/hBN heterostructures (and micro-PL for the WS₂ device) would be helpful to confirm the quality.
3. The authors claim high success rate. I think it would also be fair to show some examples (in Supplementary) of those cases which fail.
4. Line 43: $E[k]$ is incorrect – the k is the running index, so E is the function of p and N , that is $E[p,N]$.
5. Line 68: the very recent paper on the programmed assembly, published in Nature (doi:10.1038/nature23905), should be acknowledged.
6. Line 98: Was the annealing done in a separate glove box (there is no furnace in figure S1)?
7. Line 114: how long was the stacking itself? What is the task breakdown for these 32 hours? Also, what was the nature of human involvement in those 6 hours?
8. Figure 2 (page 8): The meaning of numerous black arrows is unclear? Is this the stacking sequence (which is already labelled by numbers)? It might be beneficial to highlight with colour which graphene and hBN flakes were targeted for the transfer on the edge images on panel A.
9. Line 164: What is Π ?
10. Figure 3 (page 12): What's the difference between panels F and G? The line connecting the image G to the algorithm diagram seems to be pointing to the wrong block. Same for panel F. Also, the algorithm of the flake detection, presented in Figure 3A, is rather unclear. For example, was background region selected manually?
11. Line 228: D and E captions are swapped with each other.
12. Line 243: The statement "Final alignment is conducted by a human operator" in a sense invalidates the entire paper. The procedure doesn't seem to be automated if for every transfer step the manual alignment is required. The authors should comment on this!
13. Figure S9 (page 36): panels C-F are too small to see anything on them.
14. Lines 625-627: Was this procedure automatic? If yes, what was the algorithm?
15. Line 633-634: Again, was this procedure automatic? If yes, what was the algorithm?

Reviewer #2 (Remarks to the Author):

In the manuscript "Automated searching and assembly of two-dimensional crystals to build van der Waals superlattices" Masubuchi and co-workers present an almost fully automated system to find and to transfer 2D materials within a glovebox environment. The work is certainly interesting and valuable to the community as more and more groups are focusing their research on van der Waals heterostructures. The system presented in this manuscript will enable the exploration of extremely complex vdW heterostructures which would be virtually impossible to address through manual assembly.

There are some aspects of the manuscript that should be strengthened in order to make this

contribution really useful to the community:

1) A more thorough discussion about the cleanness of the heterostructures is needed. Along the manuscript there is not a single AFM image of the transferred flakes (for example). A more comprehensive study of the mobility of fabricated devices (not only one in the supporting information) would be needed to make it a very convincing statement. Moreover, other systems such as MoS₂/WS₂ or MoS₂/MoSe₂ heterostructures could be also fabricated to demonstrate a good coupling between the transferred layers through photoluminescence measurements (this might be an easier/faster way to probe how clean the interface between the layers is).

2) Also a more thorough discussion about the control over the rotational angle between flakes. From the manuscript I understand that there is no other way to find out the crystal orientation than relying on straight edges of the flakes. If this selection of flakes done by an operator? How much accuracy (and reproducibility) can be achieved?

3) An important point is that the way the manuscript is right now it cannot be useful for the community. It describes a technique that is way to difficult to be implemented anywhere. Therefore, the potential impact of this paper is very limited in its current form. The authors have done a great effort to include the references of the equipment used to implement their setup but this effort should be even more comprehensive in order to make it possible to other groups to reproduce their setup. The same would happen with the software, just explaining how the software works is not enough to guarantee that people would reproduce it in their labs. Without providing a copy of the software the impact of this work is (again) very limited.

Therefore, taking into account the prospective impact of this work in the community working on 2D materials I believe that the manuscript is worthy of publication after addressing the points above. Very specially the point 3, without providing all the details to facilitate the replication of the setup and software this work won't be useful to other groups. Providing those details I can imagine that this paper would gather a huge number of citations.

Reviewer #3 (Remarks to the Author):

Authors present an apparatus for the automatic location and (almost) automatic stacking of mechanically exfoliated flakes to form vertical heterostructures.

Heterostructures based on 2D materials are extremely important material systems both for fundamental studies and applications. Therefore the ability of fabricating complex stacks consisting of several layers with careful control of interfaces and relative angles is highly relevant.

The main novelty of the work reported can be identified in the computer vision software capable of accurately identifying single layers of different materials. The software is also capable of creating a library of flakes, which can be selected by the user to design the stack of choice. This represents a significant technical advantage as the time for searching the flakes can be relatively long, especially if the heterostructure consists of several layers.

The robotic system itself is very impressive from the technical point of view, but scientifically does not represent a novelty. Indeed, such kind of systems have been used to assemble heterostructures for long time, see for example Cao et al. Nano Lett. 15, 4914 (2015): "The relatively small size of available 2D crystals necessitates micrometer-scale precision in their positioning during transfer and encapsulation. Such positioning is impossible using commercially available gloveboxes, which permit high-resolution optical microscopy but always rely on manual operation. To this end, a fully motorized micromanipulation station has been specially built, which allows us to cleave, align, and transfer micrometer-sized crystals robotically, using translation

stages and micromanipulators, all programmed and operated remotely from the outside with joysticks." Such systems are now commonly used in different institutions, see for example the website of the Cambridge Graphene Centre (<https://www.graphene.cam.ac.uk/facilities/cleanrooms/cgc-class-1-000-cleanroom>): "Inert-atmosphere (Ar) glovebox with a robotized 3D graphene transfer system and a thermal evaporator. The thermal evaporator can be accessed from the inside as well as outside of the glovebox. There is a spinner and a hotplate inside."

Beside the novelty, there are a number of concerns that would need to be carefully addressed:

1. It is unclear why the system is fully enclosed in a glovebox if the material production -as far as one can tell from the figures- is done elsewhere and then transferred into the glovebox.

2. It is unclear how the correct answers and false detection (as reported in Figure 3P) are validated. Are the samples characterized by Raman spectroscopy or other techniques to confirm their thickness?

3. The accuracy of alignment, both in terms of shift and angle should be discussed. This is particularly important as it seems that the final alignment is done by an operator rather than the software.

4. A detailed description of the transfer process is missing. To date, several approaches for "stamping" 2D materials have been reported and from the manuscript it is not clear which approach is actually used for the assembly. The stamping process can have significant effect on the interfaces, which ultimately determines the quality of the heterostructure. Also, unwanted effects such as formation of blisters, which limits the usable area of the heterostructure, strongly depends on the process used. Elvacite is presumably melted at the end of the process?

5. The manuscript states that "the use of metal alignment marks must be avoided because it contaminates samples and restricts the thermal cleaning capability". Such a constraint would impede to directly use the heterostructure for any fabrication processes, in particular electron beam lithography, which is usually the technique of choice for patterning such small samples. Presumably it is expected that some reference marks are fabricated on the substrate containing the heterostructure, but this would require a lithography and metal deposition to be done chip by chip, rather than preparing a markers wafer beforehand.

6. The main concern of this reviewer is that no characterization of the interface between the 2D materials is provided. At least, in particular Raman and AFM map should be included to validate the quality of the heterostructures formed using the system reported. Cleanliness of the interface is typically the most critical point of any heterostructure, and bubbles and blisters are quite common and requires care

The formation of blisters or contamination, usually happening at the interfaces between 2D materials, should also be commented.

7. Finally, Authors should comment regarding the possibility of using the system for the assembly of heterostructures based on large area materials (such as those grown by CVD). Indeed, any realistic use of heterostructures will require a scalable method for their assembly.

Response to reviewers:

We thank all the reviewers for their valuable comments. They helped us to make the manuscript clearer and stronger. Point by point response to reviewers' comments follows (Reviewers' comments and questions are in blue colored italic font, our responses are in black colored font, and the corrections are in red colored italic font).

Reviewers' comments:

Reviewer #1 (Remarks to the Author):

Masubuchi et al report on the development of a novel automated protocol and apparatus for micromechanical stacking of van der Waals heterostructures. The work is of high quality, definitely novel and very timely, which allows me to recommend the paper for publication in Nature Communication after the following comments are addressed:

We thank the reviewer for noting the importance of this works as “*definitely novel and very timely*”. As the reviewer points out, we consider that automating the stacking process of van der Waals (vdW) heterostructure is a fundamental step needed to exploit the full potential of 2D materials.

1 The details of the heterostructure fabrication are rather vague.

As the reviewer points out, the details of the heterostructure fabrication was missing in the previous version of the manuscript. To complement the information, we added the schematic flow diagram of fabrication process in Fig. 1, focusing on how we handled the polymer stamp. Based on Fig. 1, we address the queries raised by the reviewer as follows;

Fig. 1 The schematic flow diagram of the fabrication process, including the pre- and post-treatments of the SiO₂/Si

substrates. The process consists of (A) Exfoliation, (B) Searching, (C) Annealing, (D) Assembly, and (E) Drop off. Among them, (A) and (E) were conducted in the ambient air. The processes (B) through (D) were conducted in the glovebox enclosure.

How often the polymer stamp has to be replaced?

A polymer stamp can be utilized throughout the fabrication process of a vdW heterostructure [Fig. 1]. At the final transfer process of the vdW heterostructure onto the SiO₂/Si substrate, the polymer stamp is melted [Fig. 1E]. Thus, the polymer stamp is disposed.

How was it done?

The stamping apparatus is equipped with the vacuum chucking stage of the glass slide. The replacement was conducted by shutting the vacuum pumping line, changing the glass slide, and re-opening the pumping line. This process was done manually.

How was the final transfer onto the silicon substrate made? Was it automated as well?

The final transfer onto the silicon substrate was done using the similar stamping apparatus in the ambient air [Fig. 1E]. This process was conducted manually. Before entering the details of it, we explain why we selected this strategy. To avoid the contamination trapped between the SiO₂/Si substrate and vdW heterostructure, we needed to minimize the exposure of the SiO₂/Si substrate to the ambient. When the vdW heterostructure was transferred after long exposure of the SiO₂/Si to the ambient air (10-15 min) after the final cleaning, the transport characteristics of the hBN/monolayer graphene/hBN device (assembled in the ambient) [Fig. 2(A)] exhibited rather broad peak width (~3 V) in the R_{xx} vs. V_g curve as well as hysteresis in the V_g sweep [Fig. 2(B)]. This result indicated that the surface of SiO₂/Si substrate was contaminated during the exposure. On the other hand, when we transfer the vdW heterostructure immediately after cleaning, the hBN/trilayer graphene/hBN device (assembled in the ambient) exhibited the unprecedented high quality [e.g. Y. Asakawa, S. Masubuchi, *et al.*, PRL **119**, 186802 (2017)], and no hysteresis in the V_g sweep. From these results, we selected to conduct the final transfer in the ambient air. Otherwise, when we try to transfer the vdW heterostructure inside the glovebox, the SiO₂/Si needs to be exposed to the ambient air a bit long time (5-10 min). By taking this strategy, the fabricated device hBN/monolayer graphene/hBN (assembled in glovebox enclosure) [Fig. 2(C)] exhibited rather small peak width (<0.5 V), and no hysteresis in the V_g sweep [Fig. 2(D)], thus demonstrating the high-quality of the device.

To clarify the above points, we added the following description to the method section as follows;

LL. 306-317: After finishing the vdW heterostructures assembly, the glass slide was unloaded from the stamping apparatus, taken out of the glovebox [Fig. S5]. The final transfer was conducted by using the similar stamping apparatus developed in the ambient air. The SiO₂/Si chip was cleaned by the piranha solution (H₂SO₄:H₂O₂ = 2:1), followed by the ultrapure water rinse. The SiO₂/Si was loaded to the stamping apparatus and was heated to 185 °C. The glass slide was set to the stamping apparatus. The position where the vdW heterostructure was dropped was selected by using the XY stage. The SiO₂/Si chip was slowly lifted, and contacted to the vdW heterostructure. Then the polymer stamp was melted, thereby transferring the vdW heterostructure to the SiO₂/Si chip. After a short duration of ~20 seconds, the SiO₂/Si chip

was lowered. The polymer stamp was prolonged and torn. Finally, the SiO₂/Si chip was unloaded from the stamping apparatus and the stamp was dissolved by immersing SiO₂/Si chip in chloroform for 1 min.

Fig. 2 (A and C) The optical microscope images of the hBN/graphene/hBN devices. (A) The SiO₂ was exposed to the ambient air for long time (10-15 min) after cleaning, whereas (C) the duration was minimized (less than 1 min). (B and D) The longitudinal resistance R_{xx} as a function of back-gate bias voltage V_g measured at $T = 4.2$ K for the devices in A and C, respectively. The arrows indicate the sweep direction of V_g .

Also, how to reliably control that the stack can be peeled onto the Si substrate and not stuck on the polymer stamp?

Whether the stack is peeled to the SiO₂/Si substrate or stuck on the polymer stamp is controlled by the substrate temperature. A threshold temperature is determined by the glass transition temperature of the polymer utilized; $T_g = 90$ °C for the case of Elvacite 2552C [Technical Data Sheet of Elvacite 2552C]. When the substrate temperature is lower than the glass transition temperature $T = 80$ °C < T_g , the stack adheres to the polymer stamp. When the substrate temperature is higher than the glass transition temperature $T = 185$ °C > T_g , the polymer stamp is melted and the stack is transferred to the SiO₂/Si substrate.

To clarify the above points, we added the comments to the supplementary information in LL. 505-512.

1.1 How often was the human intervention needed during the van der Waals assembly?

In the current form of the software implementation, the 1-2 minutes of human intervention is needed in an interval of 10-15 minutes *i.e.* every cycle of pick-up. The breakdown of the 1-2 minutes human intervention is as follows; When the automated alignment is completed, the computer program pops up the notification to the human operator. The human operator checks the result of automated alignment. If needed, the operator overrides the result by moving the XY stage. When the alignment is completed, click the button. After that, the computer program automatically raises and lower the SiO₂/Si substrate thereby picking up the targeted 2D crystal.

To clarify the above points, modified the main manuscript at: LL 246-249 as follows;

The human operator is requested to confirm the alignment position before the process proceeds further. If needed,

the operator can override the alignment and perform fine tuning using the alignment assistance tools presented on the computer screen [Supplementary information].

2 *How clean are the interfaces between the 2D crystals in the fabricated stacks? AFM images for graphene/hBN heterostructures (and micro-PL for the WS₂ device) would be helpful to confirm the quality.*

To address the queries by the reviewer, we fabricated the three vdW heterostructures; hBN/4L-graphene/hBN [Fig. 3A], hBN/graphene/4-5L hBN/trilayer graphene/hBN [Fig. 3B], and hBN/MoS₂/WS₂/hBN [Fig. 3C]. The AFM images of the regions surrounded by the black dashed rectangles in Fig. 3A-C were presented in Fig. 3D-E.

In hBN/4L-graphene/hBN [Fig. 3A], the channel region was free of bubbles [white dashed rectangle in Fig. 3D]. In hBN/graphene/4-5L hBN/trilayer graphene/hBN [Fig. 3B], some bubbles were discerned [Fig. 3 E], however sufficiently large area $> 2 \times 2 \mu\text{m}$ was still remained intact, and can be provided for the tunneling transport studies. In the case of hBN/MoS₂/WS₂/hBN [Fig. 3C], the situation is like the hBN/graphene/4-5L hBN/trilayer graphene/hBN. There were some bubbles, but the sufficiently large area was remained intact [Fig. 3F].

To confirm the quality of interfaces, we measured the photoluminescence spectra in the hBN/WS₂/MoS₂/hBN heterostructure [Fig. 4]. The photoluminescence spectra were taken by using a 533-nm laser to excite the monolayers of MoS₂, WS₂ and MoS₂/WS₂ heterostructure. The laser beam was focused to a spot size of $\sim 1 \mu\text{m}$ with a total power of 1 mW. A monochromator and a Peltier-cooled charge-coupled device (CCD) were used to record the photoluminescence spectra. The photoluminescence spectra measured at monolayers of MoS₂ (green) and WS₂ (blue curve) were shown in Fig. 4. They exhibited narrow peak structure with half width at half maximum of $\Delta\lambda = 14 \text{ nm}$ for monolayer MoS₂ and $\Delta\lambda = 6.6 \text{ nm}$ for monolayer WS₂, which is comparable to the best quality device reported to date [F. Cadiz *et al.*, PRX **7**, 021026 (2017)]. In addition, in the spectrum taken at MoS₂/WS₂ heterostructure (red curve), the intensity of photoluminescence from the monolayer MoS₂ and WS₂ bands $\lambda = 664$ and 624 nm , respectively, were significantly suppressed, which was previously reported in CVD-grown MoS₂/WS₂ [Y. Gong *et al.*, Nature Materials **13**, 1135 (2014)]. This observation demonstrates the cleanness of the interface between MoS₂/WS₂.

To clarify the above points, we added the above description to the supplementary information in LL. 579-602.

Fig. 3 (A-C) Optical microscope images of (A) the tetra layer graphene encapsulated in hBN, (B) hBN/graphene/4-5L hBN/trilayer graphene/hBN, and (C) hBN/MoS₂/WS₂/hBN. (D-F) AFM images of the regions surrounded by the black dashed rectangles in (A-C).

Fig. 4 Photoluminescence spectra measured in the regions of MoS₂ (green), WS₂ (blue), and WS₂/MoS₂ (red). The correspondence to the measurement points were indicated by the colored circles in Fig. 3C.

3 *The authors claim high success rate. I think it would also be fair to show some examples (in Supplementary) of those cases which fail.*

Yes, we agree that it is fair to show some examples of the cases which fail. The most common result when we fail is that the targeted 2D crystals are remained on the surface of SiO₂/Si substrate after contacting the host 2D crystal to

the targeted 2D crystal, as schematically illustrated in Fig. 5. To illustrate some example of the failure, we show the several targeted 2D crystals that could not be picked up in Fig. 6A-C. In all cases, the host crystals were the single crystals of hBN. In the case of Fig. 6A, the thick region of hBN [white arrow in Fig. 6A] was picked-up whereas the thin hBN [red arrow in Fig. 6A] was remained on the substrate. In the case of Fig. 6B, the similar failure took place. Only the thick dot [white arrow in Fig. 6B] was lifted but the other area was remained on the substrate. As a reason of these failures, we speculate that the thick 2D crystal prevented targeted 2D crystal to be contacted to the host 2D crystals, thereby remaining it on the SiO₂/Si substrate.

In case of Fig. 6C, the trilayer graphene flake [red arrow in Fig. 6C] was not lifted. In this case, there were no thick regions in the flake, but the targeted flake was kept in the ambient air for more than 1 month before annealed in the Ar/H₂ gas. Therefore, the surface of targeted 2D crystals were presumably be contaminated.

Nevertheless, if we load the 2D crystals immediately after the exfoliation, perform annealing inside the glove box, and select the 2D crystals so that they were well separated from the other 2D crystals, we achieved almost perfect success rate. In the previous setup built in the ambient air, the success rate was gradually decreased during the transfer process, and after the several hours of the exposure of the targeted 2D crystal to the ambient air, the success rate was decreased to almost zero, which is presumably due to the contamination accumulated on the surface of the targeted 2D crystal during the fabrication process.

We speculate that the reason of high success rate realized in this system owes to the fact that the most time-consuming part of the fabrication process (searching and transfer) were conducted entirely inside the glove box enclosure, which prevents contaminating the surface.

To clarify the above points, we added the above description to the supplementary information in LL. 782-815.

Fig. 5 The schematic illustration of the typical cases which we fail.

Fig. 6 Optical microscope images of the 2D flakes that we failed to pick-up. The materials were (A) thin hBN, (B) hybrid monolayer and trilayer graphene, and (C) trilayer graphene. The trilayer graphene (C) was exposed to the ambient air for more than a month before annealing in Ar/H₂ gas.

4 Line 43: $E[k]$ is incorrect – the k is the running index, so E is the function of p and N , that is $E[p,N]$.

We thank the reviewer for this comment. We corrected the $E[k]$ to $E[p,N]$.

Line 38: The notation $E[k]$ was corrected to $E[p, N]$

5 Line 68: the very recent paper on the programmed assembly, published in Nature (doi:10.1038/nature23905), should be

acknowledged

We thank the reviewer for this comment. To cite the programmed assembly, published in Nature (doi:10.1038/nature23905), we modified the manuscript as follows;

Line 62-64: To date, some technological components for automation have been proposed, such as image analysis algorithm for segmenting graphene flakes on SiO₂/Si²⁷, and layer by layer stacking techniques of CVD-grown 2D crystals²⁸.

6 *Line 98: Was the annealing done in a separate glove box (there is no furnace in figure S1)?*

Yes, as pointed out by the referee, the annealing was performed in the quartz furnace built inside the second glovebox connected to the main glovebox. The samples were transferred to the stamping apparatus without exposing to the ambient air. To clarify this point, we modified the main text as follows;

Line 95-96: The SiO₂/Si chips were annealed in the Ar/H₂ atmosphere at 600 °C for 3 hours in the quartz furnace built inside the secondary glovebox connected to the main glovebox.

7 *Line 114: how long was the stacking itself?*

The stacking process of the vdW superlattice structures presented in the Fig. 2B was done within 8 hours.

7.1 *What is the task breakdown for these 32 hours?*

The task breakdown for the 32 hours was presented in Table 1;

To clarify this point, we added the Table 1 to the supplementary information in LL. 818-820.

Task	time
Search graphene and record their positions	8 hours
Search hBN and record their positions	8 hours
Design heterostructure	1 hour
Anneal the sample in Ar/H₂ atmosphere	4 hours
Transferring SiO₂/Si chips between experimental apparatus (e.g. transfer silicon chips between chiptray and annealing chamber)	1 hour
Assembly	8 hours
Miscellaneous tasks such as setting the slide glass, starting up stamping program, operating valves and gas handling systems for annealing apparatus... and so on.	2 hours
Total	32 hours

Table 1. The breakdown for the operation time.

7.2 *Also, what was the nature of human involvement in those 6 hours?*

The breakdown for the human involvement was presented in Table 2;

To clarify this point, we added the Table 2 in the supplementary information at LL. 821-822.

Task	
Loading silicon chips to the glovebox enclosure	10 minutes

Tuning the parameters for detecting graphene and start program	20 minutes
Tuning the parameters for detecting hBN and start program	20 minutes
Loading silicon chips for annealing chamber	20 minutes
Unloading silicon chips from annealing chamber	20 minutes
Designing vdW heterostructures	60 minutes
Tuning the parameters for assembly and start program	30 minutes
Confirm the alignment and perform manual alignment when it's needed	2x30=60 minutes
Dropping vdW stack onto the SiO₂/Si substrate	30 min.
Miscellaneous tasks; e.g. setting the slide glass, starting up computer, operating valves...	1.5 hours
Total	6 hours

Table 2. The task breakdown for the human involvement.

8 *Figure 2 (page 8): The meaning of numerous black arrows is unclear?*

8.1 *Is this the stacking sequence (which is already labelled by numbers)?*

We thank the reviewer for this comment. As pointed out by the reviewer, the black arrows indicated the stacking sequence, which was already been labelled by numbers. For the clarity, we removed the black arrows from Figure 2 in the main text.

Figure 2 (page8): black arrows were removed.

8.2 *It might be beneficial to highlight with colour which graphene and hBN flakes were targeted for the transfer on the edge images on panel A.*

We thank the reviewer for this comment. We highlighted them with colors, respectively.

Figure 2 (page8): The edge images of graphene and hBN flakes targeted for the transfer were colored.

9 *Line 164: What is P_i ?*

P_i was the normalized gray value histogram of the optical microscope image for each region. i is an index for the gray value which runs from 0 to 255.

To clarify this point, we modified the manuscript as follows:

Line 166-167: We then calculated the entropy $H = -\sum_{i=0}^{255} P_i \log_2 P_i$ of the grayscale image for each region, where P_i is the gray value histogram of the pixels which belong to the region.

10 *Figure 3 (page 12):*

10.1 *What's the difference between panels F and G? The line connecting the image G to the algorithm diagram seems to be pointing to the wrong block. Same for panel F.*

We thank the reviewer for this comment. The lines from the images F and G, which were the results of the entropy thresholding and the color thresholding, respectively, were connected correctly to the algorithm diagrams. To explain the

difference between the images F and G, we summarized the responses of these thresholding to the monolayer graphene, the bilayer graphene, corrugated crystals, and the tape residues in Table 3. In general, the entropy thresholding gives positive response to the flat objects, which is surrounded by the edges. Therefore, it gives positive response to the monolayer graphene, bilayer graphene [Table 3 and Figure 3F]. On the other hand, the color thresholding extracts the region where the optical intensity is within the set range. Therefore, it gives positive response to monolayer graphene and corrugated crystals [Table 3 and Figure 3G]. Although the difference between extracted images appears slight, however by taking the intersection between the images F and G, only the region of monolayer graphene was extracted, thereby demonstrating the different nature in these thresholding algorithms.

To clarify the above points, we added the above description to the supplementary information at ll. 824-837.

	Entropy thresholding [Fig. 3 F]	Color thresholding [Fig. 3G]	The intersection between F and G
Monolayer graphene	Positive	Positive	Positive
Bilayer graphene	Positive	Negative	Negative
Corrugated crystals	Negative	Positive	Negative
Tape residues	Positive/Negative	Negative	Negative

Table 3. The typical response of the entropy and color thresholding to the monolayer graphene, bilayer graphene, corrugated crystals, and tape residues.

10.2 *Also, the algorithm of the flake detection, presented in Figure 3A, is rather unclear. For example, was background region selected manually?*

We thank the reviewer for this comment. To fully describe the algorithm, we appended the source code of the flake detection algorithm in a supplementary file for review process [FlakeDetectionForMLG.cpp]. The functions `tdmms_finder_initialize` and `tdmms_finder_find` provides the full descriptions of the flake description algorithm. The function contains callouts for HALCON operator. The reference to the HALCON operators can be obtained at <http://www.mvtec.com/doc/halcon/13/en/>.

Selection of the background region, pointed out by the reviewer, was done manually while tuning the detection parameters.

In addition, to provide the full description on the algorithms other than the flake detection, we will make the source code publically available under an open source license in the code repository GitHub. When this manuscript has been accepted for publication, we will post the source code to the public repository <https://www.github.com/smasubuchi/tdmms/>. By taking this method, the researchers interested in this work can download software source code from the GitHub and they can compile the source code on their own computers. We believe that this is the best practice for accelerating the research activity of the vdW heterostructures. In addition, by taking this practice, the other researchers who are interested in this work can join the code sharing community, share their developments on the functionalities, image processing algorithms, and bug fixes. Finally, we would like to ask the researchers who are benefited from the software system to cite this paper.

To clarify the above points, we appended the C++ program in the supplementary file: FlakeDetectionForMLG.cpp. In addition, we added the comments on the source code availability in supplementary information at LL. 476-481.

Comments on the selecting background region was also provided in the main text at ll. 153-154 as follows; “the background image of SiO₂/Si without 2D crystals (recorded manually before starting the automated search),”

11 *Line 228: D and E captions are swapped with each other.*

We thank the reviewer for this comment. The D and E captions were swapped with each other.

Line 231: D and E captions are swapped with each other

12 *Line 243: The statement “Final alignment is conducted by a human operator” in a sense invalidates the entire paper: The procedure doesn’t seem to be automated if for every transfer step the manual alignment is required. The authors should comment on this!*

As the reviewer commented, the statement “Final alignment is conducted by a human operator” was rather vague. The concrete meaning of the above message was as follows; **“When the automated alignment process is completed, targeted 2D crystal is brought to the center of optical microscope within lateral error of 10 μm. The human operator is requested to confirm the alignment position before the process proceeds further. If it’s needed, the human operators can manually override the alignment and perform the fine tuning using the alignment assistance tools presented on the computer screen [details are presented in the supplementary information]”**

Nevertheless, as we have mentioned in the main manuscript, the automated alignment process can bring the targeted 2D crystals within a lateral error of 10 μm, and if this order of tolerance can be afforded, the system can stack the 2D crystals in a fully automated way. Indeed, the stamping demonstration presented in supplementary movie S4 was conducted by just clicking the confirmation button at the time of 00:53, without performing the manual alignment.

In addition, we intentionally implemented the system so that the human operators can override the alignment process even during the automated lifting process of SiO₂/Si chip. We believe that this strategy is favorable for good balancing between the flexibility and the compliance to the computer design. We consider that the fully automated alignment is an optional functionality, which will be implemented in the future version of the software system.

To clear the concern about the ambiguity, we modified the main text as follows:

ll. 244-249: “When the automated alignment process is completed [Fig. 5C], the targeted 2D crystal is brought to the center of optical microscope within lateral error of 10 μm [Fig. 5F–I]. The human operator is requested to confirm the alignment position before the process proceeds further. If it’s needed, the operator can manually override the alignment and perform fine tuning using the alignment assistance tools presented on the computer screen [Supplementary information].”

We added the following comments in the supplementary information at LL. 840-849:

Remarks on the final alignment process.

When we performed the automated alignment process, it can bring the targeted 2D crystals within a lateral error of 10 μm, and if this order of tolerance can be afforded, the system can stack the 2D crystals in a fully automated way. The stamping demonstration presented in supplementary movie S4 was conducted by just clicking the confirmation button at 00:53 and no manual alignment was performed. In addition, we intentionally implemented the system so that the human operators can override the alignment process even during the automated lifting process of SiO₂/Si chip. We believe that this strategy is favorable for good balancing between

the flexibility and the compliance to the computer design, and for the device production in the fundamental scientific research activity. We consider that the fully automated alignment is an optional functionality, which will be implemented in the future version of the software system.

13 *Figure S9 (page 36): panels C-F are too small to see anything on them.*

We thank the reviewer for this comment, we enlarged panels C-F in Figure S9.

Figure S14 (renumbered) (page 46): We enlarged panels C-F

14 *Lines 625-627: Was this procedure automatic? If yes, what was the algorithm?*

The edge patterns were automatically extracted by applying the Canny's edge detection algorithm [R. Szeliski *Computer Vision: Algorithms and Applications*. Springer Science & Business Media] to the optical microscope image. The conversion from the detected edge to the vector drawing format was also done computationally. In other words, the human operator is not required to manually trace the positions of the targeted 2D crystals on the computer screen.

15 *Line 633-634: Again, was this procedure automatic? If yes, what was the algorithm?*

At the current stage of implementation, final adjustment of the overlaid edge patterns to the optical microscope images was done manually. The automated alignment was limited to Line 241-243 in the main manuscript; “By performing an additional fine-tuning algorithm [Fig. 5C], the targeted 2D crystal is adjusted to the designated position within a lateral error of 10 μm [Fig. 5F–I].” We think that the implementation of the fine automated alignment algorithm based on the extracted edge patterns will be addressed in the future version of the software system.

To clear the above the above concerns, we modified the descriptions in the supplementary information as follows:

ll. 733-734: Note that these procedures were implemented in the software system and thus can be utilized by a few clicks on the computer screen.

ll. 737-738: In the current form of the software implementation, this alignment was conducted by the operator.

Reviewer #2 (Remarks to the Author):

In the manuscript "Automated searching and assembly of two-dimensional crystals to build van der Waals superlattices" Masubuchi and co-workers present an almost fully automated system to find and to transfer 2D materials within a glovebox environment. The work is certainly interesting a valuable to the community as more and more groups are focusing their research on van der Waals heterostructures. The system presented in this manuscript will enable the exploration of extremely complex vdW heterostructures which would be virtually impossible to address through manual assembly.

We thank the reviewer for noting our work as “*The work is certainly interesting a valuable to the community as more and more groups are focusing their research on van der Waals heterostructures.*”. We hope this contribution significantly accelerates the exploration for the physical properties of vdW heterostructures.

There are some aspects of the manuscript that should be strengthen in order to make this contribution really useful to the community:

1 A more thorough discussion about the cleanness of the heterostructures is needed.

1.1 Along the manuscript there is not a single AFM image of the transferred flakes (for example).

To address the queries by the reviewer, we fabricated the three vdW heterostructures; hBN/4L-graphene/hBN [Fig. 7A], hBN/graphene/4-5L hBN/trilayer graphene/hBN [Fig. 7B], and hBN/MoS₂/WS₂/hBN [Fig. 7C]. The AFM images of the regions surrounded by the black dashed rectangles in Fig. 7A-C were presented in Fig. 7D-E.

In hBN/4L-graphene/hBN [Fig. 7A], the channel region was free of bubbles [white dashed rectangle in Fig. 7D]. In hBN/graphene/4-5L hBN/trilayer graphene/hBN [Fig. 7B], some bubbles were discerned [Fig. 7E], however sufficiently large area $> 2 \times 2 \mu\text{m}$ was still remained intact, and can be provided for the tunneling transport studies. In the case of hBN/MoS₂/WS₂/hBN [Fig. 7C], the situation is like the hBN/graphene/4-5L hBN/trilayer graphene/hBN. There were some bubbles, but the sufficiently large area was remained intact [Fig. 7F].

To clarify the above points, we added the above description to the supplementary information at ll. 604-618:

Fig. 7 (A-C) Optical microscope images of (A) the tetra layer graphene encapsulated in hBN, (B) hBN/graphene/4-5L hBN/trilayer graphene/hBN, and (C) hBN/MoS₂/WS₂/hBN. (D-F) AFM images of the regions surrounded by the black dashed rectangles in (A-C).

1.2 A more comprehensive study of the mobility of fabricated devices (not only one in the supporting information) would be needed to make it a very convincing statement.

We thank the reviewer for this comment. To address the queries by the reviewer, we performed transport measurements in two hBN/graphene/hBN heterostructures at $T = 4.2$ K, which we call the device I and II, respectively. The optical microscope images of the devices I and II are shown in Fig. 8 A and E, respectively. In both devices, the longitudinal resistance (R_{xx}) as a function of back-gate bias voltage (V_g) exhibited sharp peak structure with small full width at half maximum of $\delta V_g = 0.25$ V and 0.5 V, indicating the small charge carrier inhomogeneity [Fig. 8 B and F]. The R_{xx} was converted to the longitudinal resistivity using $\rho_{xx} = \frac{W}{L} R_{xx}$, where W and L are the channel width and length. The longitudinal conductivity $\sigma_{xx} = 1/\rho_{xx}$ as a function of V_g were shown in Fig. 8 C and G. From σ_{xx} , we calculated the charge carrier mobility using the Drude model $\mu = \sigma/ne$, where n is the charge carrier density. The value of n was calculated by using $n = C_g(V_g - V_{\text{Dirac}})$, where $C_g = 1 \times 10^{-4}$ F/m² is the gate capacitance, and V_{Dirac} is the position of charge neutrality point. In the entire range of V_g , the values of charge carrier mobility exceeded $\mu > 10^6$ cm²/Vs, indicating the high quality of our devices.

To clarify the above points, we added the above descriptions to the supplementary information at ll. 558-577.

Fig. 8 (A and E) Optical microscope images of monolayer graphene encapsulated in hBN. (B and F) Longitudinal resistance (R_{xx}) as a function of back-gate bias voltage (V_g) measured at $T = 4.2$ K. (C and G) Longitudinal conductivity (σ_{xx}) as a function of V_g . (D and H) Charge carrier mobility μ as a function of V_g . The data in A-D (E-H) were measured in device I (II).

1.3 Moreover, other systems such as MoS_2/WS_2 or $\text{MoS}_2/\text{MoSe}_2$ heterostructures could be also fabricated to demonstrate a good coupling between the transferred layers through photoluminescence measurements (this might be an easier/faster way to probe how clean the interface between the layers is).

We thank the reviewer for this valuable comment. To address the queries by the reviewer, we assembled the exfoliated hBN, WS_2 , MoS_2 , and hBN flakes [Fig. 9A], and fabricated hBN/ WS_2 / MoS_2 /hBN heterostructure [Fig. 9B]. The AFM image in the region surrounded by the black dashed square in Fig. 9B is shown in Fig. 9C.

The photoluminescence spectra were taken by using a 533-nm laser to excite the monolayers of MoS_2 , WS_2 and MoS_2/WS_2 heterostructure. The laser beam was focused to a spot size of $\sim 1 \mu\text{m}$ with a total power of 1 mW. A monochromator and a Peltier-cooled charge-coupled device (CCD) were used to record the photoluminescence spectra. The photoluminescence spectra measured at monolayers of MoS_2 (green) and WS_2 (blue curve) were shown in Fig. 9D. They exhibited narrow peak structure with half width at half maximum of $\Delta\lambda = 14$ nm for monolayer MoS_2 and $\Delta\lambda = 6.6$ nm for monolayer WS_2 , which is comparable to the best quality device reported to date [F. Cadiz *et al.*, PRX **7**, 021026 (2017)]. In addition, in the spectrum taken at MoS_2/WS_2 heterostructure (red curve), the intensity of photoluminescence from the monolayer MoS_2 and WS_2 bands $\lambda = 664$ and 624 nm, respectively, were significantly suppressed, which was previously reported in CVD-grown MoS_2/WS_2 [Y. Gong *et al.*, Nature Materials **13**, 1135 (2014)]. This observation demonstrates the cleanness of the interface between MoS_2/WS_2 .

To clear the concerns, we added the above description to the supplementary information in ll. 579-602:

Fig. 9 (A) Optical microscope images of the hBN, WS₂, MoS₂, and hBN flakes (top to bottom) utilized for assembling WS₂/MoS₂ heterostructures. (B) Optical microscope image of the hBN/WS₂/MoS₂/hBN heterostructure. The scale bar corresponds to 5 μm. (C) Atomic force microscope image of the region surrounded by the black dashed square in B. (D) Photoluminescence spectrum measured in the regions of WS₂ (green), MoS₂ (blue), and WS₂/MoS₂ (red). The correspondence to the measurement points were indicated by the colored circles in B.

2 *Also, a more thorough discussion about the control over the rotational angle between flakes. From the manuscript, I understand that there is no other way to find out the crystal orientation than relying on straight edges of the flakes.*

If this selection of flakes done by an operator?

As the reviewer pointed out, the crystal orientation was inferred relying on the straight edges of the 2D crystals. The selection of the flakes was performed by an operator. To illustrate the fitting process, we present the screenshots of the CAD software [Fig. 10]. Fig. 10A shows the optical microscope image of the exfoliated hBN flake. The edge patterns extracted from Fig. 10A are presented in Fig. 10B. The operator draws a box on the computer screen [white arrow in Fig. 10B]. Then the straight line is fitted to the edge pattern by using the least-squares method [Fig. 10C]. The extracted angle of the straight line to the horizontal axis was (i) $\theta = 17.4215^\circ$. The other edges can also be fitted as (ii) $\theta = 77.2986^\circ$, and (iii) $\theta = 17.5254^\circ$ [Fig. 10D]. The relative angle between (i) and (ii) was $\Delta\theta_{i-ii} \sim 59.87^\circ$, whereas that between (i) and (iii) was $\Delta\theta_{i-iii} = 0.1^\circ$. Note that, these values reflect the crystallographic symmetry of hBN, where the expected exfoliation angles were $\Delta\theta_{i-ii} = 60^\circ$ and $\Delta\theta_{i-iii} = 0^\circ$. From these results, we estimate the extent of error in the fitting process to be $\sim 0.1^\circ$.

To address the above points, we added the above description to the supplementary information at ll. 850-869.

Fig. 10 The demonstrations of the edge fitting. The presented images are the screenshots of the software. (A) The representative thick hBN flake. (B) (white curves) the edge patterns extracted from the image A. The rectangle highlighted by the white arrow indicates the region that are going to be fitted by the straight line. (C) The extracted angle between the horizontal line was 17.4215° . (D) The extracted angles of the other edges relative to the horizontal line were 77.2986° and 17.5254° .

How much accuracy (and reproducibility) can be achieved?

To illustrate the accuracy and reproducibility in the rotational angle between flakes, we provide in Fig. 11, the comparison between the designed (A and B) and obtained angles (C and D) for the two representative devices numbered 0110 and 0112. In the device 0110, the alignment angles between the set of straight edges were designed as 3.0° [Fig. 11A]. The measured angles between the corresponding straight edges in the fabricated device were 2.57° [Fig. 11B]. In the device 0112, the alignment angles between the set of edges were designed as 40.3° [Fig. 11C] and the result of measurement was 39.0° [Fig. 11D]. However, note that the fitting to the straight edges in the optical microscope images were conducted by hand-fitting. From these data, the accuracy of the alignment angle can be estimated to be less than $< \pm 1.5^\circ$

To address the above points, we added the above description to the supplementary information at ll. 871-885.

Fig. 11 A and B, the relative angles between the straight edges in the vdW heterostructures design. C and D the optical microscope images of the fabricated vdW heterostructures. Note that, in these devices, the lateral positions were intentionally shifted with respect to the designed position during the assembly. Therefore, the alignment error in the lateral direction needs to be ignored.

3 *An important point is that the way the manuscript is right now it cannot be useful for the community. It describes a technique that is way too difficult to be implemented anywhere. Therefore, the potential impact of this paper is very limited in its current form. The authors have done a great effort to include the references of the equipment used to implement their setup but this effort should be even more comprehensive to make it possible to other groups to reproduce their setup.*

We completely agree with the reviewer's concern that providing sufficient information for the replication is a key factor for accelerating the research activity. To clear the reviewers' concern, we present in Fig. 12 the full connectivity diagram between the hardware components utilized in the system. This schematic describes the complete logical connection between the hardware components. Therefore, by replicating the hardware connection and install the software codes, which will be publically available under an open source license, other groups will be able to reproduce these setups. However, at the same time, we must admit that providing the full and concrete description of the hardware system is virtually impossible because the system includes many hand-assembled parts, and the system is currently under further developments to include the automated exfoliation functionality. Therefore, we would like to ask the researchers who are interested in the replication of this system to contact the author directly. We are willing to help them replicate our

system.

The information presented below is presented in the supplementary information at ll. 458-467.

Fig. 12 The connectivity diagram of the hardware components utilized in the system.

4 *The same would happen with the software, just explaining how the software works is not enough to guarantee that people would reproduce it in their labs. Without providing a copy of the software the impact of this work is (again) very limited.*

We completely agree with the reviewer's opinion. To clear the reviewer's concerns, we decided to make the entire software source code available under an open source license. When this manuscript has been accepted for publication, we will post the source code to the open source software repository (GitHub) at <https://github.com/smasubuchi/tdmms/>. The researchers interested in this work can download the source code from the GitHub and they can compile the code on their own computers. Furthermore, they can develop their own functionalities upon our software code when it is needed. By using the social networking system on GitHub, we can continuously share bug fixes, and work together to improve software systems. Therefore, would like to invite the other researchers to join the software development community and to work together to improve the software system that is needed for the vdW heterostructure research. We believe this is the best practice for accelerating the scientific research activity. We deserve this contribution to significantly boost the exploration activities of the vdW heterostructures. Finally, we would like to ask the researchers who are benefited from this contribution to cite this paper.

The descriptions on the source code availability was added in the main text. Ll. 288-290.

In addition, the comments to the open source development is added in the supplementary information in 476-481. Therefore, taking into account the prospective impact of this work in the community working on 2D materials I believe that the manuscript is worthy of publication after addressing the points above. Very specially the point 3, without providing all the details to facilitate the replication of the setup and software this work won't be useful to other groups. Providing those details, I can imagine that this paper would gather a huge number of citations.

We thank the reviewer for this comment. We hope this contribution to significantly benefit those researchers who are involved in vdW heterostructures research.

Reviewer #3 (Remarks to the Author):

Authors present an apparatus for the automatic location and (almost) automatic stacking of mechanically exfoliated flakes to form vertical heterostructures. Heterostructures based on 2D materials are extremely important material systems both for fundamental studies and applications. Therefore, the ability of fabricating complex stacks consisting of several layers with careful control of interfaces and relative angles is highly relevant.

The main novelty of the work reported can be identified in the computer vision software capable of accurately identifying single layers of different materials. The software is also capable of creating a library of flakes, which can be selected by the user to design the stack of choice. This represents a significant technical advantage as the time for searching the flakes can be relatively long, especially if the heterostructure consists of several layers.

We thank the reviewer for noting our work as “This represents a significant technical advantage as the time for searching the flakes can be relatively long, especially if the heterostructure consists of several layers”. We believe that this work significantly reduces the time effort that have been spent for searching the flakes. Therefore, this work frees up the researchers from the repetitive tasks, thereby letting them focusing on the scientific research activities.

The robotic system itself is very impressive from the technical point of view, but scientifically does not represents a novelty. Indeed, such kind of systems have been used to assemble heterostructures for long time, see for example Cao et al. Nano Lett. 15, 4914 (2015): "The relatively small size of available 2D crystals necessitates micrometer-scale precision in their positioning during transfer and encapsulation. Such positioning is impossible using commercially available gloveboxes, which permit high-resolution optical microscopy but always rely on manual operation. To this end, a fully motorized micromanipulation station has been specially built, which allows us to cleave, align, and transfer micrometer-sized crystals robotically, using translation stages and micromanipulators, all programmed and operated remotely from the outside with joysticks." Such systems are now commonly used in different institutions, see for example the website of the Cambridge Graphene Centre (<https://www.graphene.cam.ac.uk/facilities/cleanrooms/cgc-class-1000-cleanroom>): "Inert-atmosphere (Ar) glovebox with a robotized 3D graphene transfer system and a thermal evaporator. The thermal evaporator can be accessed from the inside as well as outside of the glovebox. There is a spinner and a hotplate inside."

As the reviewer point out, the paper by Cao *et al.*, Nano Lett. **15**, 4914 (2015) and the homepage of Cambridge Graphene Centre comments on the robotic apparatus for assembling vdW heterostructures inside the glovebox chamber. We admit that our hardware system has some aspects in common. However, we think that there is a big difference between this work and the system in the Cambridge graphene center. The difference is whether the presence of the “autonomy”, *i.e.* the computer perceives the environment and control the motors. The detailed feature comparison table is presented in Table 4. In our system, the hardware components are controlled autonomously by the software system, whereas those of the system at the Cambridge graphene center are operated remotely using the joysticks by the human operator. Therefore, the difference is like those between the self-driving car *vs.* the conventional manually controlled car, the autopilot aircraft *vs.* the manually operated aircraft, and the autonomous robot cleaner *vs.* the conventional vacuum cleaner. Therefore, we believe that this work contains sufficient technological and scientific leap from the conventional system and suffices the publication criteria of the Nature Communications.

	The system at GRC	Our system
Glovebox enclosure	✓	✓
Searching	Manual operation	Automated
Assembly	Manual operation	Semi-automated
Exchanging Si chips	Manual operation	Automated
Catalog database of 2D flakes	---	✓
CAD System	---	✓

Table 4. The feature compariton table between the system at Cambridge graphene center and the our system.

Beside the novelty, there a number of concerns that would need to be carefully addressed:

1 *It is unclear why the system is fully enclosed in a glovebox if the material production -as far as one can tell from the figures- is done elsewhere and then transferred into the glovebox.*

As the reviewer pointed out, the mechanical exfoliation of hBN and graphene were done in the ambient condition and loaded into the glovebox. The reason why we took this strategy was because of the process efficiency rather than the technological problem. The exfoliation inside the glovebox enclosure is performable, and indeed the hBN/MoS₂/WS₂/hBN device presented in Fig. 13 were fabricated by exfoliating MoS₂ and WS₂ flakes inside the glovebox enclosure. Technological merit of developing searching and assembly apparatus inside is that by conducting the most time-consuming part of the device assembly inside the glovebox enclosure, the 2D crystals remain pickable for sufficiently long time than the case when they were kept in the ambient air. If the heterostructure fabrication has been conducted in the ambient condition, the surface of the targeted 2D crystals are gradually contaminated and becomes unpickable after the several hours.

To clarify these points, we added the above information to the supplementary information as a joint correspondence to the queries raised by the reviewer#2. LL. 782-815

Fig. 13 (A) Optical microscope images of the hBN, WS₂, MoS₂, and hBN flakes (top to bottom) utilized for assembling WS₂/MoS₂ heterostructures. (B) Optical microscope image of the hBN/WS₂/MoS₂/hBN heterostructure. The scale bar corresponds to 5 μ m. (C) Atomic force microscope image of the region surrounded by the black dashed square in B. (D) Photoluminescence spectrum measured in the regions of WS₂ (blue), MoS₂ (green), and WS₂/MoS₂ (red). The correspondence to the measurement points were indicated by the colored circles in B.

2 *It is unclear how the correct answers and false detection (as reported in Figure 3P) are validated. Are the samples characterized by Raman spectroscopy or other techniques to confirm their thickness?*

First, the correlation between the optical color contrast and the layer thickness has been verified by using the transport and photoluminescence measurements. Fig. 14 shows the quantum Hall effects measured in (A) monolayer, (B) bilayer, and (C) trilayer graphene. Fig. 15 shows the Raman spectrum of monolayer, bilayer, and trilayer MoS₂. Furthermore, the thickness of SiO₂ was controlled to $290 \pm 5\%$, and the light illumination condition was kept constant by using the halogen lamp with the feedback control [LA-150FBU, Hayashi Watch Works]. Therefore, the correlation between the optical microscope images and the layer thickness is guaranteed. Upon these basis, the false detection rate was manually aggregated by checking all the extracted optical microscope images. Table 5 shows the spreadsheet summary utilized for calculating the false detection rate. Note that the confusion between layer thickness is relatively small and the major source of false detection derives from the contaminating objects such as particles [Fig. 16].

To clear the reader's concerns, the information presented above is added to the supplementary information in II. 902-926.

Fig. 14 (A)-(C) Optical microscope images of (A) monolayer, (B) bilayer, and (C) trilayer graphene devices. The scale bars correspond to 4 μm . (D)-(F) Color plots of the longitudinal resistance R_{xx} measured as a function of back-gate bias voltage V_g and perpendicular magnetic fields B at $T = 1.7$ K for (A) monolayer, (B) bilayer, and (C) trilayer graphene devices.

Fig. 15 A and B Optical microscope images of monolayer, bilayer, and trilayer MoS₂. Raman spectrum of MoS₂ flakes measured for (A) monolayer, (B) bilayer, and (C) trilayer by using 533-nm laser.

Material	# of chips inspected	Elapsed time (h)	Detected	# of false positive (by layer thickness)	# of false positive (by particle)	Percentage of correct answer
Monolayer Graphene	27	4	1886	8	115	93.5
Bilayer Graphene	27	4	1668	8	25	98.0
Trilayer Graphene	27	4	1129	31	7	96.6
Monolayer MoS ₂	31	6	362	4	23	92.5
Bilayer MoS ₂	31	6	166	4	1	97.0
Trilayer MoS ₂	31	6	288	3	1	98.6

Table 5. Spreadsheet summary of the figures utilized for extracting the false detection rate.

Fig. 16 The example of false detection.

- 3 *The accuracy of alignment, both in terms of shift and angle should be discussed. This is particularly important as it seems that the final alignment is done by an operator rather than the software.*

The accuracy of alignment in terms of angle

To illustrate the accuracy and reproducibility in the rotational angle between flakes, we provide in Fig. 17, the comparison between the designed (A and B) and obtained angles (C and D) for the two representative devices numbered 0110 and 0112. In the device 0110, the alignment angles between the set of straight edges were designed as 3.0° [Fig. 17A]. The measured angles between the corresponding straight edges in the fabricated device were 2.57° [Fig. 17B]. In the device 0112, the alignment angles between the set of edges were designed as 40.3° [Fig. 17C] and the result of measurement was 39.0° [Fig. 17D]. However, note that the fitting to the straight edges in the optical microscope images were conducted by hand-fitting. From these data, the accuracy of the alignment angle can be estimated to be less than $< \pm 1.5^\circ$.

To address the above points, we added the above description to the supplementary information at ll. 871-885.

Fig. 17 A and B, the relative angles between the straight edges in the vdW heterostructures design. C and D the optical microscope images of the fabricated vdW heterostructures. Note that, in these devices, the lateral positions were intentionally shifted with respect to the designed position during the assembly. Therefore, the alignment error in the lateral direction needs be ignored.

The accuracy of alignment in terms of translational shift

To demonstrate the of alignment in terms of the translational shift, we conducted the alignment demonstration as presented in Fig. 18. Fig. 18A-H shows the image of the hBN crystal in the CAD software screen. In these designs, the alignment center was locked to the fixed position on the hBN flake, as indicated by the red lines in Fig. 18A-H. To demonstrate the alignment accuracy, each design was rotated by the step of $\theta = 45^\circ$. The result of automated alignment in the stamping apparatus were presented in Fig. 18I-P. Note that no human intervention was made during the alignment. The targeted flake was aligned to the designated position within a lateral error of $10 \mu\text{m}$.

To address the above points, we added the description to the supplementary information at ll. 887-900.

Fig. 18 The demonstration of the alignment. (A-H) The optical microscope image of the hBN crystal in the CAD screen. The hBN flake was rotated by a step of $\theta = 45^\circ$. The targeted alignment center was indicated by the red straight lines. (I-P) The optical microscope image after alignment. The screen center was indicated by the red straight lines.

4 *A detailed description of the transfer process is missing. To date, several approaches for "stamping" 2D materials have been reported and from the manuscript is not clear which approach is actually used for the assembly. The stamping process can have significant effect on the interfaces, which ultimately determines the quality of the heterostructure. Also, unwanted effects such as formation of blisters, which limits the usable area of the heterostructure, strongly depends on the process used. Elvacite is presumably melted at the end of the process?*

We thank the reviewer for this comment. To complement the information, we added the pictorial flow diagram on the overall process of the device fabrication, especially focusing on how we handled the polymer stamp in Fig. 19. After finishing the vdW heterostructures assembly, the glass slide was unloaded from the stamping apparatus, took out of the glovebox, and brought to the similar stamping apparatus developed in the ambient air [Fig. 1(d)]. The SiO₂/Si chip was cleaned by the piranha solution (H₂SO₄:H₂O₂ = 2:1), followed by the ultrapure water rinse. The SiO₂/Si was loaded to the stamping apparatus and was heated to 185 °C. The glass slide was set to the stamping apparatus. The position (translational direction) between the vdW stack and the SiO₂/Si chip was aligned by using the XY stage. Then, the SiO₂/Si chip was slowly lifted, and contacted to the vdW heterostructure. The polymer stamp was melted, thereby transferring the vdW heterostructure to the SiO₂/Si chip. After a short duration (about 20 seconds), the SiO₂/Si chip was lowered. The polymer stamp was prolonged and torn. Finally, the SiO₂/Si chip was unloaded from the stamping apparatus and was immersed in chloroform for 1 min.

To address the above points, we added the overall pictorial flow diagram in ll. 494-503. The methods section in the main text was modified as follows; After finishing the vdW heterostructures assembly, the glass slide was unloaded from the stamping apparatus, taken out of the glovebox [Fig. S5]. The final transfer was conducted by using the similar stamping apparatus developed in the ambient air. The SiO₂/Si chip was cleaned by the piranha

solution ($H_2SO_4:H_2O_2 = 2:1$), followed by the ultrapure water rinse. The SiO_2/Si was loaded to the stamping apparatus and was heated to 185 °C. The glass slide was set to the stamping apparatus. The position where the vdW heterostructure was dropped was selected by using the XY stage. The SiO_2/Si chip was slowly lifted, and contacted to the vdW heterostructure. Then the polymer stamp was melted, thereby transferring the vdW heterostructure to the SiO_2/Si chip. After a short duration of ~20 seconds, the SiO_2/Si chip was lowered. The polymer stamp was prolonged and torn. Finally, the SiO_2/Si chip was unloaded from the stamping apparatus and the stamp was dissolved by immersing SiO_2/Si chip in chloroform for 1 min.

Fig. 19 The schematic flow diagram of the fabrication process, including the pre- and post-treatments of the SiO_2/Si substrates. The process consists of (A) Exfoliation, (B) Searching, (C) Annealing, (D) Assembly, and (E) Drop off. Among them, (A) and (E) were conducted in the ambient air. The processes (B)-(D) were conducted in the glovebox enclosure.

5 The manuscript states that "the use of metal alignment marks must be avoided because it contaminates samples and restricts the thermal cleaning capability". Such a constrain would impede to directly use the heterostructure for any fabrication processes, in particular electron beam lithography, which is usually the technique of choice for patterning such small samples. Presumably it is expected that some reference marks are fabricated on the substrate containing the heterostructure, but this would require a lithography and metal deposition to be done chip by chip, rather than preparing a markers wafer beforehand.

We thank the reviewer for this comment. Basically, we consider that this comment is based on misunderstanding. By using our protocol, the final transfer of vdW heterostructure can be made on arbitrary substrates. Therefore, by transferring the vdW heterostructure onto the SiO_2/Si substrate with metal alignment marks, we could perform further

processing. Indeed, the devices with metal contacts for transport studies presented in the manuscript has been processed by using the electron beam lithography. The statement “the use of metal alignment marks must be avoided because it contaminates samples and restricts the thermal cleaning capability” applies only to the starting 2D crystals exfoliated on the SiO₂/Si substrate.

To avoid the confusion by the reader, we modified the main text as follows;

Line 133-135: The use of metal alignment marks must be avoided for the SiO₂/Si substrates on which the starting 2D crystals were exfoliated because it contaminates surface and restricts the thermal cleaning capability. On the other hand, the fabricated vdW heterostructure should be transferred onto the SiO₂/Si with metal alignment marks for further processing.

6 *The main concern of this reviewer is that no characterization of the interface between the 2D materials is provided. At least, in particular Raman and AFM map should be included to validate the quality of the heterostructures formed using the system reported. Cleanliness of the interface is typically the most critical point of any heterostructure, and bubbles and blisters is quite common and requires care. The formation of blisters of contamination, usually happening at the interfaces between 2D materials, should also be commented.*

To address the queries by the reviewer, we fabricated the three vdW heterostructures; hBN/4L-graphene/hBN [Fig. 3A], hBN/graphene/4-5L hBN/trilayer graphene/hBN [Fig. 19B], and hBN/MoS₂/WS₂/hBN [Fig. 19C]. The AFM images of the regions surrounded by the black dashed rectangles in Fig. 3A-C were presented in Fig. 3D-E.

In hBN/4L-graphene/hBN [Fig. 19A], the channel region was free of bubbles [white dashed rectangle in Fig. 19D]. In hBN/graphene/4-5L hBN/trilayer graphene/hBN [Fig. 19B], some bubbles were discerned [Fig. 19E], however sufficiently large area $> 2 \times 2 \mu\text{m}$ was still remained intact, and can be provided for the tunneling transport studies. In the case of hBN/MoS₂/WS₂/hBN [Fig. 19C], the situation is like the hBN/graphene/4-5L hBN/trilayer graphene/hBN. There were some bubbles but the sufficiently large area was remained intact [Fig. 19F].

To confirm the quality of interfaces, we measured the photoluminescence spectra in the hBN/WS₂/MoS₂/hBN heterostructure [Fig. 21]. The photoluminescence spectra were taken by using a 533-nm laser to excite the monolayers of MoS₂, WS₂ and MoS₂/WS₂ heterostructure. The laser beam was focused to a spot size of $\sim 1 \mu\text{m}$ with a total power of 1 mW. A monochromator and a Peltier-cooled charge-coupled device (CCD) were used to record the photoluminescence spectra. The photoluminescence spectra measured at monolayers of MoS₂ (green) and WS₂ (blue curve) were shown in Fig. 21. They exhibited narrow peak structure with half width at half maximum of $\Delta\lambda = 14 \text{ nm}$ for monolayer MoS₂ and $\Delta\lambda = 6.6 \text{ nm}$ for monolayer WS₂, which is comparable to the best quality device reported to date [F. Cadiz *et al.*, PRX 7, 021026 (2017)]. In addition, in the spectrum taken at MoS₂/WS₂ heterostructure (red curve), the intensity of photoluminescence from the monolayer MoS₂ and WS₂ bands $\lambda = 664$ and 624 nm , respectively, were significantly suppressed, which was previously reported in CVD-grown MoS₂/WS₂ [Y. Gong *et al.*, Nature Materials 13, 1135 (2014)]. This observation demonstrates the cleanness of the interface between MoS₂/WS₂.

To clear the concerns, we added the above description to the supplementary information in ll. 579-602.

Fig. 20 (A-C) Optical microscope images of (A) the tetra layer graphene encapsulated in hBN, (B) hBN/graphene/4-5L hBN/trilayer graphene/hBN, and (C) hBN/MoS₂/WS₂/hBN. (D-F) AFM images of the regions surrounded by the black dashed rectangles in (A-C).

Fig. 21 Photoluminescence spectra measured in the regions of MoS₂ (green), WS₂ (blue), and WS₂/MoS₂ (red). The correspondence to the measurement points were indicated by the colored circles in Fig. 3C.

7 *Finally, Authors should comment regarding the possibility of using the system for the assembly of heterostructures based on large area materials (such as those grown by CVD). Indeed, any realistic use of heterostructures will require a scalable method for their assembly*

We thank the reviewer for this comment. Yes, this system is applicable to the assembly of heterostructures based

on CVD grown 2D crystals. Indeed, we have succeeded in picking up the CVD-grown 2D crystals by using the same stamping procedure in the ambient condition [Y. Hoshi *et al*, PRB 95, 241403 (2017)].

The comments on this work was added in the supplementary information at LL. 928-933.

For all the reviewers:

Finally, to meet the format requirements of the nature communications, we have shortened the abstract to be less than 150 words:

Van der Waals (vdW) heterostructures are comprised of stacked atomically thin two-dimensional (2D) crystals and serve as novel materials providing unprecedented physical properties. However, since the initial fabrication of vdW heterostructures, the random natures in positions and shapes of exfoliated 2D crystals have required the repetitive manual tasks of optical microscopy-based searching and mechanical transferring, thereby severely limiting the complexity of heterostructures. We therefore developed a robotic system that automatically searches exfoliated 2D crystals and assembles them into superlattices inside the glovebox. Its enduring capacity was shown to be the detecting of 400 monolayer graphene flakes per hour with a small error rate (< 7%) and stacking of four cycles of the designated 2D crystals per hour. The system enabled fabrication of the superlattice consisting of 29 alternating layers of the graphene and the hexagonal boron nitride. This capacity provides a scalable approach for prototyping a variety of vdW superlattices.

Accordingly, the start of introduction was modified as follows:

LL. 25-31

The family of exfoliatable and functional 2D crystals¹ is rapidly growing, exhibiting various electronic properties such as ferromagnets^{2, 3}, semiconductors⁴, superconductors⁵, and topological insulators⁶. Recent advancements of 2D crystal research were enabled by key technological breakthroughs, including mechanical exfoliation⁷ and the assembly of vdW heterostructures^{1, 7, 8}. The precise interface controllability and wide choice of materials with various electronic properties enable vdW heterostructures to have high technological potentials that cannot be achieved by conventional semiconductor heterostructures^{9, 10, 11}.

REVIEWERS' COMMENTS:

Reviewer #1 (Remarks to the Author):

The authors have answered all the questions and have amended the manuscript as requested. Since the human intervention was required during each transfer step, the title "Automatic searching and assembly..." is misleading and should be corrected. Human intervention should be explicitly mentioned in the abstract. In the discussion/conclusions section, the directions to improve the procedure towards more autonomous assembly should be provided (with necessary references). I believe this work is the first big step towards the true automatic fabrication of 2D heterostructure and therefore deserves to be published in Nature Communications.

Reviewer #2 (Remarks to the Author):

The authors have done a very thorough work to reply all the points raised by this (and as far as I can assess the other referees). I highly valued the efforts done by the authors to provide as much details as possible to facilitate the replication of the system by other groups. Therefore I would recommend the manuscript for its publication in Nature Communications after minor considerations.

The authors clarified some points in their response that indicates that the automated system has some limitations regarding the accuracy of the sliding translation, rotation alignment and the interface cleanliness. A reproducibility of 10 microns in lateral displacement is a severe limitation of the technique, the rotational alignment accuracy of $\pm 1^\circ$ is not best than the reported results in the literature with manually actuated systems and the presence of multiple bubbles leaving clean areas of only 2×2 microns is not ideal for device fabrication. Therefore, I believe that these limitations of the current setup (as well as the outlook and further future improvements) should be discussed in the conclusions or in the wrapping up of the manuscript.

Reviewer #3 (Remarks to the Author):

Authors have carefully and clearly responded to all the points raised by the reviewers.

The revised version further confirms that the major novelty of work is in the computer vision software capable of automatically map samples and accurately identifying single layers of different materials, creating a "library" of flakes to be used for heterostructures.

The automatic mapping and the overall apparatus are very impressive, however the assembly process seems far from being automated, indeed:

- both production of flakes and stamping onto the final substrate is performed manually outside of the glovebox
- stamping outside of the glovebox is essential (Fig. 2)
- human intervention is required at every cycle of pick up
- the lateral error is $10 \mu\text{m}$, comparable with the size of the flakes (especially single layers). Manual fine tuning is likely to be required frequently.
- Fully automated alignment is considered only as "an optional functionality, which will be implemented in the future"

Beside the automation being only partial, I am also concerned about the quality of the interfaces produced. The concentration of bubbles at the interfaces is extremely high and very far from the state of the art (see for example Pizzocchero et al., Nat Commun 7, 11894), leaving only small region actually usable. Presumably the devices showing high mobility have been fabricated in

these specific areas. Also, figure 21 only refers to a 1umx1um spot, whereas a map should have been provided to prove good quality interfaces across the entire heterostructure.

Response to reviewers:

We thank all the reviewers for understanding the importance of this work and recommending this manuscript for publication in Nature Communications. Their comments helped us to make the manuscript clearer and stronger. Point by point response to reviewers' comments follows (Reviewers' comments and questions are in blue colored italic font, our responses are in black colored font, and the corrections are in red colored italic font).

Reviewers' comments:

Reviewer #1 (Remarks to the Author):

The authors have answered all the questions and have amended the manuscript as requested. Since the human intervention was required during each transfer step, the title "Automatic searching and assembly..." is misleading and should be corrected. Human intervention should be explicitly mentioned in the abstract. In the discussion/conclusions section, the directions to improve the procedure towards more autonomous assembly should be provided (with necessary references). I believe this work is the first big step towards the true automatic fabrication of 2D heterostructure and therefore deserves to be published in Nature Communications.

We thank the reviewer for noting this work as “the first big step towards the true automatic fabrication of 2D heterostructures” and recommending this manuscript for publication in Nature Communications.

By following the suggestions given by the referee, to avoid the misleading by the readers, we propose to replace “automated” by “Autonomous robotic”. In the standard textbook in this field, “*Autonomous Robots: From Biological Inspiration to Implementation and Control* George, A. Bekey, MIT Press”, the definition of autonomy is described as follows; “*Autonomy refers to systems capable of operating in the real-world environment without any form of external control for extended periods time (page 1)*”. In addition, the scope of the Springer journal “*Autonomous Robots*” (<https://link.springer.com/journal/10514>) is described as “*Autonomous Robots reports on the theory and applications of robotic systems capable of some degree of self-sufficiency*”. In the operation of our system, flake detection, automated alignment, and 2D crystal location algorithms perceives the environment such as optical microscope images and operate the robots for extended periods time without external control. These features suffice the use of term “autonomous” in the time of this manuscript.

Therefore, we propose the title of this manuscript as “*Autonomous robotic searching and assembly of two-dimensional crystals to build van der Waals superlattices*”

In addition, to explicitly mention the human intervention, we modified the abstract as follows;

Van der Waals heterostructures are comprised of stacked atomically thin two-dimensional crystals and serve as novel materials providing unprecedented physical properties. However, the random natures in positions and shapes of exfoliated two-dimensional crystals have required the repetitive manual tasks of optical microscopy-based searching and mechanical

transferring, thereby severely limiting the complexity of heterostructures. To solve the problem, here we develop a robotic system that searches exfoliated two-dimensional crystals and assembles them into superlattices inside the glovebox. The system can autonomously detect 400 monolayer graphene flakes per hour with a small error rate (< 7%) and stack four cycles of the designated two-dimensional crystals per hour with few minutes of human intervention for each cycle. The system enabled fabrication of the superlattice consisting of 29 alternating layers of the graphene and the hexagonal boron nitride. This capacity provides a scalable approach for prototyping a variety of van der Waals superlattices.

Reviewer #2 (Remarks to the Author):

The authors have done a very thorough work to reply all the points raised by this (and as far as I can assess the other referees). I highly valued the efforts done by the authors to provide as much details as possible to facilitate the replication of the system by other groups. Therefore I would recommend the manuscript for its publication in Nature Communications after minor considerations.

We thank the reviewer for noting our efforts for providing as much details as possible to facilitate the replication of the system and recommending this manuscript for publication in Nature Communications.

The authors clarified some points in their response that indicates that the automated system has some limitations regarding the accuracy of the sliding translation, rotation alignment and the interface cleanliness. A reproducibility of 10 microns in lateral displacement is a severe limitation of the technique, the rotational alignment accuracy of ± 1 degree is not best than the reported results in the literature with manually actuated systems and the presence of multiple bubbles leaving clean areas of only 2x2 microns is not ideal for device fabrication. Therefore, I believe that these limitations of the current setup (as well as the outlook and further future improvements) should be discussed in the conclusions or in the wrapping up of the manuscript.

Yes, as the reviewer points out, it is better to add discussions in the limitations of the current setup as well as the outlook and further future improvements in the wrapping up of the manuscript.

Here, we added the following discussions in the main text in ll. 411-452.

The fabrications of the vdW superlattice structures presented in Fig. 2 and Fig. 3 were achieved by implementing the above technologies. Here, we would like to discuss the current limitations of the setup as well as the outlook for the future improvements. These can be split into three technological groups; the stamping process, the alignment, and the human intervention. First, in the stamping process, the formation of blisters between 2D crystals are not fully avoided [Supplementary Note 4]. To solve the problem, the improvements in the stamping process, such as the hot pick-up method³⁶, needs to be implemented. Second, the final alignment accuracy gained after using the alignment assistance tool becomes less than $\pm 1^\circ$ and $\pm 1 \mu\text{m}$, respectively [Fig. 7]. However, to achieve this accuracy, manual alignment process is needed [Supplementary Notes 8, 9, 10, and 11]. This process needs to be automated in the future developments of the image detection algorithm and error handling schemes. Third, for the human intervention, the exfoliation and final transfer was

performed manually outside the glovebox. To automate these procedures, robotic systems for exfoliation, wafer cleaning, and Si chiptray transfer between the components, etc. will be needed. The recent rapid developments in the machine learning and computer vision algorithms might help realizing these technologies.

In summary, we developed the autonomous robotic system for searching 2D crystals and assembling them into vdW heterostructures. Because this system is contained in a glovebox, the system can handle oxygen- and humidity-sensitive 2D crystals, such as black phosphorous^{20,21} and niobium diselenide²². In addition, the system is in principle applicable to the assembly of heterostructures from CVD-grown 2D crystals [Supplementary Note12]. The wider material design freedom enabled by our system offers unprecedented opportunities for exploring the full potential of vdW heterostructures. By this development, we can reduce the human intervention involved in the vdW heterostructure fabrication by orders. We believe that this work free up researchers from repetitive tasks and letting them to focus on more intellectually creative tasks. Therefore, this is a fundamental step forward to realizing the dreamscape of artificial materials by vdW heterostructures. With respect to the materials aspect of vdW heterostructures, the digital nature of the materials properties of 2D crystals and the wide choice of combinations for vdW heterostructures have good compatibility with the information-processing algorithms of machine-learning and Bayesian statics³⁷, which will open new trajectories of combinatorial materials research on vdW superlattices.

Reviewer #3 (Remarks to the Author):

Authors have carefully and clearly responded to all the points raised by the reviewers.

The revised version further confirms that the major novelty of work is in the computer vision software capable of automatically map samples and accurately identifying single layers of different materials, creating a "library" of flakes to be used for heterostructures.

The automatic mapping and the overall apparatus are very impressive; however, the assembly process seems far from being automated, indeed:

- both production of flakes and stamping onto the final substrate is performed manually outside of the glovebox*
- stamping outside of the glovebox is essential (Fig. 2)*
- human intervention is required at every cycle of pick up*
- the lateral error is 10um, comparable with the size of the flakes (especially single layers). Manual fine tuning is likely to be required frequently.*
- Fully automated alignment is considered only as "an optional functionality, which will be implemented in the future"*

We thank the reviewer for carefully reading our manuscript and noting that “the automated mapping and the overall apparatus are very impressive”. At the same time, we note that there remains some process that were done manually inside and outside the glovebox enclosure such as mechanical exfoliation. To answer the reviewer’s concerns, we would like to comment on the future perspectives for resolving the points raised by the reviewer.

- both production of flakes and stamping onto the final substrate is performed manually outside of the glovebox

In the current setup, this limitation derives from the limited cleaning capability of Si substrate inside the glovebox. This could be resolved by installing the oxygen plasma etching apparatus inside the glove box.

- stamping outside of the glovebox is essential (Fig. 2)

This comment seems to be based on the misunderstanding of the manuscript. The stamping for fabricating the heterostructure in Fig. 2 was done inside the glovebox. Final drop was done manually outside the glovebox.

- human intervention is required at every cycle of pick up

- the lateral error is 10um, comparable with the size of the flakes (especially single layers). Manual fine tuning is likely to be required frequently.

- Fully automated alignment is considered only as "an optional functionality, which will be implemented in the future"

Yes, these matters need to be solved in the future developments. However, we would like to comment that these problems can be solved by implementing the software algorithms without changing the hardware apparatus. Therefore, these issues does pose strong constraints on the future development capability of the system. The future advancements in the software system will be shared with the other researches through the GitHub community.

Beside the automation being only partial, I am also concerned about the quality of the interfaces produced. The concentration of bubbles at the interfaces is extremely high and very far from the state of the art (see for example Pizzocchero et al., Nat Commun 7, 11894), leaving only small region usable. Presumably the devices showing high mobility have been fabricated in these specific areas. Also, figure 21 only refers to a 1umx1um spot, whereas a map should have been provided to prove good quality interfaces across the entire heterostructure

We thank the reviewer for drawing our attention to the manuscript for bubble-free stamping method presented in Pizzocchero et al., Nat Commun 7, 11894. Indeed, we expect that the formation of blisters can be resolved by performing the stamping at elevated temperature $T = 110^{\circ}\text{C}$. We commented on the possibility for resolving the blisters in the discussion section, as well as the reference to the paper.

The reviewer also comments that an atomic force microscope map should have been provided in Figure 21. We cannot provide an atomic force microscopy map because the devices were already mounted in the ceramic package. However, all the spin and valley degeneracies were resolved in the moderate magnetic fields $B < 4 \text{ T}$, indicating high quality of our device. If the blisters were included in the channel region, we cannot expect such high-quality behavior. Therefore, we think that, at least, the channel region was free of blisters.